

# Secondary organic aerosol formation in biomass-burning plumes: Theoretical analysis of lab studies and ambient plumes

Qijing Bian[1], Shantanu H. Jathar[2], John K. Kodros[1], Kelley C. Barsanti[3], Lindsay E. Hatch[3], Andrew A May[4], Sonia M. Kreidenweis[1], and Jeffrey R. Pierce[1,5]

[1]Department of Atmospheric Science, Colorado State University, Fort Collins, CO, USA
[2]Department of Mechanical Engineering, Colorado State University, Fort Collins, CO, USA
[3]Department of Chemical and Environmental Engineering and College of Engineering – Center for Environmental Research and Technology (CE-CERT), University of California, Riverside, CA, USA
[4]Department of Civil, Environmental and Geodetic Engineering, the Ohio State University, Columbus, OH, USA
[5]Department of Physics and Atmospheric Science, Dalhousie University, Halifax, NS, Canada

## Abstract

Secondary organic aerosol (SOA) has been shown to form in biomass-burning emissions in laboratory and field studies. However, there is significant variability among studies in mass enhancement, which could be due to differences in fuels, fire conditions, dilution, and/or limitations of laboratory experiments and observations. This study focuses on understanding processes affecting biomass-burning SOA formation in laboratory smog-chamber experiments and in ambient plumes. Vapor wall losses have been demonstrated to be an important factor that can suppress SOA formation in laboratory studies of traditional SOA precursors; however, impacts of vapor wall losses on biomass-burning SOA have not yet been investigated. We use an aerosol microphysics model that includes representations of volatility and oxidation chemistry to estimate the influence of vapor wall loss on SOA formation observed in the FLAME-III smog-chamber studies. Our simulations with base-case assumptions for chemistry and wall loss predict a mean OA mass enhancement (the ratio of final to initial OA mass, corrected for particle-phase wall losses) of 1.8 across all experiments when vapor wall losses are modeled, roughly matching the mean observed enhancement during FLAME-III. The mean OA enhancement increases to over 3 when vapor wall losses are turned off, implying that vapor wall losses reduce the apparent SOA formation. We find that this decrease in the apparent SOA formation due to vapor wall losses is robust across the ranges of uncertainties in the key model assumptions for wall-loss and mass-transfer coefficients and chemical mechanisms.

We then apply similar assumptions regarding SOA formation chemistry and physics to smoke emitted into the atmosphere. In ambient plumes, the plume dilution rate impacts the organic partitioning between the gas and particle phases, which may impact the potential for SOA to form as well as the rate of SOA formation. We add Gaussian dispersion to our aerosol microphysical model to estimate how SOA formation may vary under different ambient-plume conditions (e.g. fire size, emission mass flux,





atmospheric stability). Smoke from small fires, such as typical prescribed burns, dilutes
      rapidly, which drives evaporation of organic vapor from the particle phase, leading to
more effective SOA formation. Emissions from large fires, such as intense wildfires,
      dilute slowly, suppressing OA evaporation and subsequent SOA formation in the near
field. We also demonstrate that different approaches to the calculation of OA
      enhancement in ambient plumes can lead to different conclusions regarding SOA
formation. OA mass enhancement ratios of around 1 calculated using an inert tracer,
      such as BC or CO, have traditionally been interpreted as exhibiting little or no SOA
formation; however, we show that SOA formation may have greatly contributed to the
      mass in these plumes.

In comparison of laboratory and plume results, the possible inconsistency of OA
      enhancement between them could be in part attributed to the effect of chamber walls
and plume dilution. Our results highlight that laboratory and field experiments that focus
      on the fuel and fire conditions also need to consider the effects of plume dilution or
vapor losses to walls.

1. Introduction

      Biomass burning is an important source of carbonaceous compounds that have
significant influence on air quality (Jaffe and Widger, 2012), climate (Bond et al., 2013)
      and human health (Naeher et al., 2007; Jassen, 2012; Johnston et al., 2012).  It is a
major source of primary fine carbonaceous (black and organic carbon) particles (Akagi
      et al., 2011), but the contribution of biomass burning to ambient concentrations of
secondary organic aerosol (SOA, organic aerosol formed in the atmosphere) is highly
      variable because of the complexities of physical and chemical evolution of biomass-
burning plumes. Laboratory studies have observed both significant organic aerosol (OA)
      increase and OA decrease in biomass-burning emissions (Hennigan et al. 2011; Ortega
et al., 2013). Some field studies of biomass burning also observed organic aerosol (OA)
      formation (Grieshop et al., 2009; Yokelson et al., 2009) and some showed little OA
production or even a net loss (Akagi et al., 2012; May et al., 2015). OA consists of
      thousands of species, but only a small portion of these have been identified, and thus
understanding of phase partitioning and the chemistry occurring in biomass-burning
      emissions is still poor (Heilman et al., 2014).

The semi-volatile nature of biomass-burning primary organic aerosol (POA) as identified
      in recent studies (Grieshop et al., 2009; May et al., 2013) further complicates the phase
dynamics during the evolution of biomass-burning emissions, both in the laboratory and
      in ambient air. In an ambient plume, positive impacts on emitted OA mass could occur
by the condensation of low-volatile organics produced from the oxidation of volatile and





semi-volatile organics (Yokelson et al., 2009); while on the other hand, reductions in OA
mass could occur due to evaporation of organic vapors driven by dilution (Jolleys et al.,
2012) or by fragmentation reactions creating higher-volatility species. Hence,
observations of OA evolution in the field are always influenced by plume dilution and
complex chemical pathways that compete for OA enhancement and loss (Akagi et al.
2012; May et al. 2015) and it is difficult to observationally separate those effects. An
extensive literature search reveals little work exploring how fire conditions (e.g. fire size
and mass flux) and atmospheric stability conditions (e.g. unstable or stable) affect OA
evolution in a chemically evolving plume and how those factors would influence the
observed plume characteristics.

To reduce some of the complexity inherent in ambient observations, smog chambers
are widely used to study the evolution of organic aerosol. The mechanism of particle
wall loss has been well studied (Crump and Seinfeld, 1981; McMurry and Rader, 1985;
Pierce et al., 2008) and is commonly used to correct aerosol measurements in smog-
chamber studies (Weitkamp et al., 2007; Hennigan et al., 2011). Wall loss of organic
vapors may also be important and leads to impacts on gas-particle partitioning in
chamber experiments, as has been demonstrated in recent studies (Matsunaga and
Ziemann, 2010; Yeh and Ziemann, 2015; Zhang et al., 2015; Bian et al., 2015;
Krechmer et al., 2016). Vapor uptake to Teflon chamber walls demonstrates absorptive
partitioning behavior following Henry's Law. The resulting loss of SOA precursors to
chamber walls makes them unavailable for reaction and leads to underestimates of
SOA production in chamber studies (Matsunaga and Ziemann, 2010; Yeh and Ziemann,
2015; Zhang et al., 2014; Zhang et al., 2015). Zhang et al. (2014) predicted that vapor
wall losses in a 25 $m^3$ chamber may lead to factor-of-4 underestimates of SOA mass
formation from biogenic and anthropogenic precursor vapors. Kokkola et al. (2014) also
showed that SOA formation from ozonolysis of α-pinene may be underestimated by a
factor up to 4 in a 4 $m^3$ chamber. Based on the work of Lim and Ziemann (2009) and
Matsunaga and Ziemann (2010), La et al. (2016) suggested that SOA yield from
mixtures of alkanes, alkenes and alcohols or ketones may be underestimated by a
factor of 2 in chambers of volumes of 5.9 and 1.7 $m^3$. Cappa et al. (2016) estimated that
SOA was increased by factors of ~2-10, depending on scenario, when vapor wall losses
were accounted for in air quality model simulations. However, it has also been pointed
out that increasing seed surface area could effectively compete for vapor absorption,
suppressing vapor wall losses and increasing SOA formation in chamber studies
(Zhang et al., 2014; McVay et al., 2014). Nah et al. (2016) also observed that the effects
of vapor wall deposition on SOA mass yields could be constrained if vapor
condensation occurs under quasi-equilibrium growth (i.e. the particles and vapors reach
equilibrium quickly).



Several modeling studies have examined SOA formation in ambient air from biomass-burning emissions (Mason et al., 2001; Alvarado and Prinn, 2009; Alvarado, et al., 2015). One difficulty is that the compounds that act as precursors of SOA in biomass-burning emissions are not well understood. Including only known SOA precursors (mainly aromatic species like toluene) in the model largely underestimates SOA production, probably because of limited knowledge about additional SOA precursor vapors, such as intermediate-volatility organic compounds (IVOCs) (Alvarado and Prinn, 2009; Jathar et al., 2014). Alvarado et al. (2015) included assumptions of unidentified IVOCs, semi-volatile and extremely low volatility organic compounds in the modeling of OA and $O_3$ formation and successfully reproduced ambient observations. However, their study did not consider the specific impacts of vapor wall losses on laboratory observations of biomass-burning SOA and how this might constrain SOA formation chemistry. Further, dilution effects on SOA formation during plume transport have not yet been investigated. In previous work, Bian et al. (2015) showed that organic-vapor wall loss in Teflon chamber experiments may drive evaporation of primary biomass-burning organic aerosol; however, the resulting impacts on SOA formation were not investigated in that work.

In this study, we (1) investigate the influence of vapor wall loss on biomass-burning SOA formation in a smog chamber, based on current knowledge of particle and vapor wall-loss rates, and (2) explore the effect of dilution on SOA formation in ambient plumes. For the smog-chamber portion of this work, we use an aerosol-microphysics model that includes particle/vapor wall losses and SOA chemistry to simulate observations reported in Hennigan et al. (2011) from smog chamber experiments conducted in the third Fire Lab At Missoula Experiments (FLAME III) study. For the ambient-plume portion of this work, we add Gaussian dispersion to the aerosol-microphysics-chemistry model, and perform sensitivity simulations that capture the effects of fire size, variable mass flux, and atmospheric stability. We describe our aerosol-microphysics model in Section 2. The smog-chamber model is described in Section 2.1, and the ambient plume model is described in Section 2.2. In Section 3.1, we present results for the sensitivity of the smog-chamber simulations to SOA-chemistry assumptions. In Section 3.2, we demonstrate the influence of vapor wall loss on SOA formation in smog-chamber experiments. In Section 3.3, we investigate the impact of fire/plume characteristics on SOA formation in ambient plumes, based on the knowledge gained from simulating the lab studies, and Section 4 presents our conclusions.

## 2. Methods

2.1. Smog-chamber simulations





Wood-smoke primary organic aerosol partitioning and SOA formation were investigated
for the smog-chamber experiments conducted during the FLAME III study from Sep-Oct
2009 (Hennigan et al., 2011; May et al., 2013 and 2014; Ortega et al., 2011; Bian et al.,
2015). Eighteen fuels that frequently burn in wild or prescribed fires across North
America were studied (Table 1). In each experiment, the combustion emissions were
introduced into the smog chamber at a dilution ratio of ~25:1 (relative to the
158 USDA/USFS Fire Sciences Laboratory, FSL, combustion chamber). Photo-oxidation
was initiated for 3-4.5 hr using sunlight / UV light after a 75 min dark period during which
primary gas and particle concentrations were characterized in the smog chamber.
Additional experimental details are included in Hennigan et al. (2011), May et al. (2013),
and Bian et al. (2015).

For our smog-chamber simulations, we use the TwO-Moment Aerosol Sectional
(TOMAS) microphysics model (Adams and Seinfeld, 2002; Pierce and Adams, 2009;
Pierce et al., 2011) combined with particle and vapor wall-loss algorithms and a SOA
production matrix to estimate SOA formation for the 18 FLAME III experiments
considered in Bian et al. (2015). Simulated aerosol species include black carbon,
organics, and water with 36 logarithmically spaced size sections from 3 nm to 10 μm. In
our previous study examining the influence of wall loss on primary semi-volatile
organics in the chamber (Bian et al., 2015), we simulated eight organic "species" within
the Volatility Basis Set (Donahue et al., 2006) with logarithmically spaced effective
saturation concentrations ($C^*$) spanning from $10^{-3}$ to $10^4$ μg m$^{-3}$ using the volatility
distribution derived by May et al. (2013). $C^*$ of $10^4$ μg m$^{-3}$ is the least-constrained
volatility bin in the analysis of May (et al., 2013), and the large amount of material in this
bin may represent some of the vapor in higher bins. In this current study, we expand the
176 simulated organics from eight to fifteen "species" including more volatile organics
between $10^6$ to $10^{11}$ μg m$^{-3}$, based on the FLAME-4 study of Hatch et al. (2016), to
178 account for chemical transformations from both volatile and semi-volatile organic
species (Fig. 1a). As described in Bian et al. (2015), we retrieved a representative
turbulence rate ($k_e$, s$^{-1}$, Crump and Seinfeld, 1981) by applying the Aerosol Parameter
Estimation (APE) model to SMPS data following the method in Pierce et al. (2008). We
then estimated the size-dependent particle wall-loss rates ($k_{w,p}(D_p)$, Eqn 1) and
reversible vapor wall-loss rate coefficients ($k_{w,on}$ and $k_{w,off}$, Eqn 2 and 3) using the fitted
turbulence rate ($k_e$),

$$k_{w,p}\left(D_p\right) = k_{w,p0} + \frac{6\sqrt{k_eD}}{\pi R}D_1\left(\frac{\pi\gamma_s}{2\sqrt{k_eD}}\right) + \frac{v_s}{4R/3} \qquad \text{Eqn 1}$$

$$k_{w,on} = \left(\frac{A}{V}\right)\frac{\left(\frac{\alpha_w\bar{c}}{4}\right)}{1.0+\left(\frac{\pi}{2}\right)\left[\frac{\alpha_w\bar{c}}{4\left(k_eD_{gas}\right)^{0.5}}\right]} \qquad \text{Eqn2}$$



$$k_{w,off} = \frac{k_{w,on}}{K_w C_w} = k_{w,on} \left( \frac{C^* M_w \gamma_w}{C_w M_p \gamma_p} \right) \qquad \text{Eqn3}$$

where $D$ is the Brownian diffusivity of the particle of size $D_p$, $R$ is the radius of the chamber, $v_s$ is the gravitational settling velocity of the particle, and $k_{w,p0}$ is a size-independent wall-loss rate that is used to represent the effect of electrostatic forces on the wall loss. $D_1$ is the Debye function (Abramowitz and Stegun, 1964). The fitted values of $k_e$ and $k_{w,p0}$ are listed in Table 1. $k_{w,on}$ is the rate coefficient for the transfer of gas-phase organic vapors to the wall, $A/V$ is the surface to volume ratio of the chamber, $\alpha_w$ is the mass accommodation coefficient of vapors onto the chamber walls, $\overline{c}$ (m s$^{-1}$) is the mean thermal speed of the molecules (calculated using the molecular weights of each organic volatility bin), $k_e$ is a function of the turbulent kinetic energy in the chamber (derived from the APE model described above), and $D_{gas}$ is the molecular diffusivity (m$^2$ s$^{-1}$). $k_{w,\,off}$ is the evaporation rate coefficient from the wall. $K_w$ is the gas-particle partitioning coefficient. $C_w$ is the equivalent or effective organic mass concentration of the walls (in units of mass per chamber volume). $C^*$ is the saturation concentration (µg m$^{-3}$). $M_p$ and $M_w$ are the average molecular weights of the organic species in the particles and in the Teflon film comprising the chamber (µg m$^{-3}$). $\gamma_w$ and $\gamma_p$ are the activity coefficients of the organic species in the Teflon film and the particle phase, respectively.

Previous studies have shown two variables primarily control vapor wall-loss rates: the effective saturation of vapor with respect to the wall ($C_w/M_w\gamma_w$) and the accommodation coefficient for vapor into the wall (Bian et al., 2015, Zhang et al., 2015). Matsunaga and Ziemann (2010) suggested $C_w/M_w\gamma_w$ values of 9, 20, 50 and 120 µmole m$^{-3}$ for n-alkanes, 1-alkenes, 2-ketones, and 2-alcohols, respectively. Krechmer et al. (2016) extended the vapor-wall-loss study of Matsunaga and Ziemann (2010) to species over a broader volatility range, suggesting that $C_w$ be treated as a function of C*. Zhang et al. (2015) also implied that $C_w$ could depend on C*, but their calculated $C_w$ values were smaller than those recommended by Krechmer et al. (2016) for $C^*$ lower than $10^5$ µg m$^{-3}$. For the mass accommodation coefficient of vapors on wall ($\alpha_w$), Matsunaga and Ziemann (2010) found it to be above $1\times10^{-5}$ while Zhang et al (2015) found that $\alpha_w$ is also dependent on $C^*$. In our simulations of the smog-chamber experiments that are presented here, we use the Krechmer $C_w/M_w\gamma_w$ values and a $\alpha_w$ of $1\times 10^{-5}$ in the base-case simulations and then perform sensitivity tests by varying $C_w/M_w\gamma_w$ and $\alpha_w$ according to the range of previously reported values.

The gas-phase organic chemistry matrix used in the model follows the study of Jathar et al. (2014). We assume that only functionalization occurs in the biomass-burning experiments, with the product organic vapors having volatilities that are either 2 or 4 volatility bins lower than the parent (Table 2). We also do not include aerosol-phase and


heterogeneous reactions in our model. SOA mass yield $\alpha_{i,j}$ is assumed to be 1 for all
reactions. We use this simple assumption of unity SOA mass yield as a first test of
226 chemistry in our chamber and plume systems as we found that we did not have enough
information to constrain the yields beyond this. The chemical mechanism is represented
as follows:

$$\frac{d[X_j]}{dt} = -k_{OH,X_j}[OH][X_j] \qquad \text{Eqn 4}$$

$$\frac{d[M_i]}{dt} = \sum_j \alpha_{i,j} k_{OH,X_j}[OH][X_j] \qquad \text{Eqn 5}$$

where $[X_j]$ represents the concentration of a gas–phase species in volatility bin j, $k_{OH,x}$ is
232 the reaction rate constant between the oxidant OH and the organic species $X_j$, and $\alpha_{i,j}$ is
the mass yield of gaseous product $M_i$ in volatility bin i (assumed to be 1 in our study).
OH exposure (OH concentration integrated over the time of the experiment) for each
experiment is taken from Hennigan et al. (2011) and the average OH exposure across
all of the experiments is assigned to the two experiments with missing values (Table 1).
OH concentration ($[OH]$) is estimated on the assumption that the photochemical aging
time in all the experiments was 4 hours. $k_{OH}$ is computed from the mathematical
relationship retrieved by Jathar et al. (2014) based on the data of Atkinson and Arey
(2003): $k_{OH} = -5.7 \times 10^{-12} \ln(C^*) + 1.14 \times 10^{-10}$ for aromatics and $k_{OH} = -1.84 \times 10^{-12}$
$\ln(C^*) + 4.27 \times 10^{-10}$ for alkanes. We use the fits for aromatics (faster chemistry) and
242 alkanes (slower chemistry) separately in different simulations to provide bounds for the
chemical reaction rates. As the relationships were derived from a limited number of
244 species, we applied a minimum $k_{OH}$ value to constrain the extrapolation to the broader
volatility range, as these relationships give negative $k_{OH}$ values at the highest volatility
bins. We then test the sensitivity of the OA enhancement ratios to the choice of
minimum $k_{OH}$ value of either $5 \times 10^{-12}$ or $1 \times 10^{-12}$. We do not consider condensed-phase
chemistry in this study. The initial values of parameters used in the model simulations,
including temperature, particle number concentration, number size distribution, mass
concentration and organic mass fraction, are listed in Table 1 for each experiment.

2.2 Investigating OA in expanding plumes

We apply a simple Gaussian-dispersion framework to represent plume volume
expansion in our box model. We assume that the pollutants are uniformly distributed
within a box with a crosswind width of $y \pm 2\sigma_y$ and height $z \pm 2\sigma_z$ (the thickness of the
box in the wind direction is fixed at 1 m), so that the plume volume in the simulation is
256 $4\sigma_y \times 4\sigma_y \times 1$ m$^3$. We assume that the initial plume width ($\sigma_y$) is the same as the fire width
(the square root of the fire area). The maximum plume height ($\sigma_z$) is constrained by the
258 boundary layer depth, which is set to be 2500 m, equivalent to a $\sigma_z$ of 625 m. We
perform sensitivity tests for fire areas of $1 \times 10^{-4}$, $1 \times 10^{-2}$, 1 and $1 \times 10^2$ km$^2$ (equivalent





initial $\sigma_y$ of 2.5, 25, 250, and 2500 m, respectively) for a neutral atmospheric stability class (D) and an emission mass flux of $5\times10^{-6}$ kg m$^{-2}$ s$^{-1}$ (on the larger end of the fluxes
in the GFED4 emission inventory as found by Sakamoto et al., 2016). The smallest fire size ($1\times10^{-4}$ km$^2$) was selected to represent a prescribed fire and the larger fire sizes (1
and $1\times10^2$ km$^2$) represent wildfire sources. For a fire size of 1 km$^2$, we also test the sensitivity to atmospheric stability class (A (unstable), D (neutral) and F (stable)) for
mass fluxes of $2\times10^{-8}$ and $5\times10^{-6}$ kg m$^{-2}$ s$^{-1}$. The dispersion parameters used to estimate $\sigma_y$ and $\sigma_z$ for different Pasquill stability classes are taken from Klug (1969). The
background is considered to be non-volatile OA with a fixed concentration of 5 µg m$^{-3}$, and this aerosol is entrained into the box as it expands. The organic-vapor chemistry
scheme is the same as used in the chamber study. The input parameters for the TOMAS Gaussian dispersion dilution simulations are listed in Table 3.

2.3 Definitions of OA enhancement

  We use two definitions of the "observed" OA enhancement ratio, both found in the
literature, to demonstrate that these definitions impact the amount of apparent SOA formation in chambers and in plumes. In smog-chamber and field studies of biomass
burning, the OA mass enhancement ratio is often calculated as the change in OA mass relative to the background, and also relative to a species assumed to be inert on the
experimental timescales. A commonly reported variable is the normalized excess mixing ratio (NEMR; Akagi et al., 2012), where the in-plume OA concentrations are corrected
for background concentrations and normalized to an inert tracer (IT) also emitted from the fire (e.g. CO or black carbon [BC]):

$NEMR_t = \dfrac{(OA_{in-plume/chamber,t} - OA_{background})}{(IT_{in-plume/chamber,t} - IT_{background})}$     Eqn 6

  where $t$ denotes that NEMR is a time-dependent (equivalently, downwind-distance-
dependent) variable. If the OA and IT are non-reactive and non-depositing (or depositing at the same rate), and OA is nonvolatile, then NEMR remains unchanged
with time and represents the emitted ratio of the two species, specific to the fuel and combustion conditions; as such, it can be compared with lab studies aimed at
quantifying these emission ratios (e.g., May et al., 2014). In the case of smog-chamber experiments, the OA and IT background concentrations are negligible because the
chamber is filled with clean air before injecting emissions. In this work, we use BC mass as our IT (Grieshop et al., 2009; Hennigan et al., 2011). We further normalize $NEMR_t$ by
the initial NEMR value (at the start of the lab experiments or at emission for the expanding plumes) to define the inert OA mass enhancement ratio (OAER$_{inert}$) (Eqn 7):

$OAER_{inert} = \dfrac{NEMR_t}{NEMR_0}$     Eqn 7





The subscript 0 refers to values at the initial time, and the subscript t refers to any subsequent time in the simulations or observations. As BC concentration decreases due to particle-phase wall losses (in smog chambers) and dilution (in ambient plumes), $OAER_{inert}$ normalizes the relative change in OA by the decrease in concentration of BC, and thus corrects for particle-phase wall losses and dilution. If these are the only processes occurring, then $OAER_{inert}$ remains fixed at a value of 1 at any time t. Other situations result in time-dependent $OAER_{inert}$. Net OA production leads to $OAER_{inert}$ values greater than 1, and net OA evaporation leads to $OAER_{inert}$ values less than 1. $OAER_{inert}$ is thus a scale factor that can be applied to OA emission factors to account for time-dependent in-plume net production/loss of OA.

Although $OAER_{inert}$ can be computed readily from observations and can indicate when other processes besides dilution are active, POA evaporation and SOA production may compensate for each other, so that it is impossible to quantify the impact of SOA production through $OAER_{inert}$ (or NEMR) alone, as has been pointed out previously (e.g., DeCarlo et al., 2010; Akagi et al. 2012; May et al., 2015). On the other hand, via the modeling approach used in this work we can directly compare simulations with and without chemistry, and thus we can isolate the impact of chemistry on our simulations and on the "observed" $OAER_{inert}$. We define the chemistry OA mass enhancement ratio ($OAER_{chem}$) as the ratio of predicted OA concentrations in the chemistry-on and chemistry-off simulations:

$$OAER_{chem} = (OA_{chem\ on,t} - OA_{background})/(OA_{chem\ off,t} - OA_{background}) \qquad \text{Eqn 8}$$

While $OAER_{chem}$ is not calculable from field or lab observations, it is the indicator of how SOA production enhances OA in the model, with all other processes being equal.

3. Results and discussions

3.1. Simulated chamber SOA production in absence of particle and vapor wall losses

This section describes simulations where we test our assumptions about gas-phase chemistry with vapor and wall losses turned off. Specifically, we test the sensitivity of OA to our assumed $k_{OH}$ values and the drop in volatility of organic product species (relative to the parent compound) with each reaction with OH (Table 2).

Fig. 2 shows the OA enhancement ratios for each of our first set of chemistry sensitivity cases. In these simulations, $OAER_{inert}$ and $OAER_{chem}$ are equivalent as chemistry is the only process affecting OA mass (no wall losses or dilution), so the OA enhancement ratios in Fig. 2 represent both OAERs described above. The starting volatility distribution in these simulations shown in Fig. 1a. Each bar in Fig. 2 is the OA enhancement ratio averaged over simulations of all 18 experiments. The predicted OA enhancements are insensitive to the chosen minimum $k_{OH}$ values (i.e. $5 \times 10^{-12}$ and $1 \times 10^{-}$



$^{12}$ cm$^3$ molec$^{-1}$ s$^{-1}$); the difference in OA enhancement ratios for these choices is less than 1%. We therefore use a minimum value of 5×10$^{-12}$ cm$^3$ molec$^{-1}$ s$^{-1}$ throughout the rest of this study. The OA enhancement ratio for the four-volatility-bin drop assumption,

Case A (1.9±0.2 for aromatic $k_{OH}$ set and 1.6±0.2 for alkane $k_{OH}$ set), is slightly larger than for the case assuming a two-volatility-bin drop, Case B (1.8±0.2 for aromatic $k_{OH}$

set and 1.5±0.2 alkane $k_{OH}$ set). The OA enhancement ratios simulated using the aromatic $k_{OH}$ set are larger than those using the alkane $k_{OH}$ set, because $k_{OH}$ for

aromatics is generally larger than alkanes when C* is lower than 10$^8$ µg m$^{-3}$. Therefore, in the remaining simulations presented here, we use the aromatic $k_{OH}$ set with a four-

volatility-bin drop per reaction as an upper bound for SOA formation, and the alkane $k_{OH}$ set with the two-volatility-bin drop per reaction as a lower bound for SOA formation.

3.2. Influence of particle and vapor wall losses on the apparent SOA production in smog chambers

This section investigates the impact of particle and vapor wall losses on the apparent SOA formation in the FLAME-III chamber studies. Fig. 3a shows the time evolution of

organic material between the gas, particle, and wall phases, when both particle and vapor wall losses are considered in the model. The first hour simulates the evolution of

primary emitted vapor and particulate organics in the dark period prior to initiating photochemistry. OM in the vapor phase decreases as vapor is absorbed into the wall.

OM in the particle phase decreases due to both direct particle losses and the loss of aerosol-phase mass from evaporation of the particles driven by the vapor losses to the

walls. The extent of the vapor wall loss is mainly controlled by the reversible vapor wall loss rate coefficients (i.e. $k_{on}$ and $k_{off}$) in Eqn 3. These two variables are mainly

influenced by two vapor-wall interaction parameters:  the effective saturation concentration of vapor with respect to the wall ($C_w/M_w\gamma_w$), and the accommodation

coefficient for vapor with the wall, $\alpha_w$ (Bian et al., 2015). We demonstrate the sensitivity of our results to values of these parameters later in this section.

The starting volatility distribution of the chemistry portion of simulations with vapor wall loss on (and base-case assumptions) is shown in Fig. 1b, representing the volatility

distribution after 1 hour of vapor-aerosol-wall re-equilibration during the "dark" phase of each smog chamber experiment (see Bian et al. (2015) for a full analysis of these

experiments). Photo-oxidation was then initiated and the simulations were continued for 4 hours. The dotted lines in Figure 3a show how the system evolves over the 5 hours of

the experiment when no photo-oxidation is allowed to occur. This evolution is contrasted with that depicted by the solid lines, for which the chemical oxidation mechanism was

activated in the model after the first hour (dark / equilibration period), to represent the experimental period when chamber irradiation began; chemistry was allowed to proceed

for the next 4 hours. In Figure 3, the upper-bound chemistry assumptions have been applied ($k_{OH}$ set for aromatics with a four-volatility-bin drop per reaction). In Figure 3a,



since particle and vapor wall losses were allowed to continue to occur in parallel with SOA formation from vapor oxidation, the extent of net SOA formation depends on the
competition between the oxidation of organic vapors and wall losses of these same vapors, as well as the competition between absorption of product vapors into the walls
and into the aerosol phase. The role of the vapors lost to the walls is explored in Fig. 3b, which shows the same case but with vapor wall losses turned off. More SOA is
produced in this second case, and OM in the vapor phase is strongly reduced due in part to the higher efficiency of the chemical reactions. In both scenarios, the produced
SOA from vapor oxidation compensates some of the OM particle wall loss, but stronger OM production also leads to more OM lost to the wall as deposited particles (green
lines). As demonstrated in these examples, the net SOA production in chambers is therefore dependent on interactions between the photochemical reaction rates (and
associated changes in organic volatility) and the wall-loss kinetics and applicable parameters (i.e., wall saturation concentration and mass accommodation coefficient of
vapors to the wall).

The $OAER_{chem}$ value for the base simulations with vapor wall losses on is 2.6±0.5 (i.e.,
the ratio of the solid red to dashed red lines in Fig. 3a, calculated by Eqn 7) after 5 hours, while the $OAER_{chem}$ value for the simulations with vapor wall losses off (Fig. 3b)
is 3.4±0.7 at this same time. Thus, these simulations suggest that vapor wall losses measurably reduce the amount of SOA formed in the chamber by removing precursor
vapors. On the other hand, the averaged $OAER_{inert}$ value (the metric used by Hennigan et al. (2011) to report their experimental observations) for our simulations with vapor
wall losses on (Fig. 3a, using BC as the tracer, not shown on this figure) is 1.9±0.4 after 5 hours, while our $OAER_{inert}$ value for the simulations with vapor wall losses off (Fig. 3b)
is 3.3±0.7. Thus, the $OAER_{inert}$ values are lower than the $OAER_{chem}$ values when vapor wall losses are on, but the two metrics are similar when vapor wall losses are off. This
difference arises because evaporation of OA, driven by vapor wall losses, decreases the OA/BC ratio throughout the experiment, lowering the value of $OAER_{inert}$. Since vapor
wall losses drive evaporation in both the chem-on and chem-off experiments, $OAER_{chem}$ is a better metric for isolating the effect of chemistry than is $OAER_{inert}$. However,
because the differences between $OAER_{inert}$ and $OAER_{chem}$ are not great when vapor wall losses are off, and because $OAER_{inert}$ is more directly comparable to the
experimental analysis of Hennigan et al. (2011), we use $OAER_{inert}$ as the representative OA enhancement ratio for the remainder of the discussion on smog-chamber SOA. We
will revisit $OAER_{chem}$ when discussing ambient plumes, where $OAER_{inert}$ and $OAER_{chem}$ show important differences.

The range of $OAER_{inert}$ values presented in Hennigan et al. (2011) was 1.7±0.7, so our comparable simulations with vapor wall loss on are in very good agreement with those
observations. Our simulations also show that these experimentally derived


enhancement ratios would be higher in the absence of vapor wall loss, since our
simulated OAER$_{inert}$ for the simulations with vapor wall losses off is almost doubled,
3.3±0.7. As the predicted underestimation of SOA formation attributed to vapor wall
losses depends on our assumptions for various wall-loss parameters and the details of
the chemistry scheme, the rest of this section explores how robust these results are to
the wall-loss and chemical-mechanism uncertainties.

We perform sensitivity tests using documented values of $C_w/M_w\gamma_w$ (9, 20, 50, 120 µg m$^{-3}$
and two sets of $C_w/M_w\gamma_w$ that vary with volatility) to estimate their influence on SOA
production in the simulated chamber experiments. $\alpha_w$ is set to 10$^{-5}$. Fig. 4a summarizes
the predicted values of OAER$_{inert}$ under our upper-bound chemistry assumptions ($k_{OH}$
set for aromatics with four-volatility-bin drop per reaction) for the various $C_w/M_w\gamma_w$
assumptions, while Fig. 4b shows the same but for the lower-bound chemistry
assumptions ($k_{OH}$ set for alkanes with two-volatility-bin drop per reaction). The OAER$_{inert}$
values using Krechmer's $C_w/M_w\gamma_w$ set are comparable to those using the fixed 9 µg m$^{-3}$
value but less than Zhang's $C_w/M_w\gamma_w$ set, because Krechmer's $C_w/M_w\gamma_w$ leads to more
vapor wall losses than Zhang's $C_w/M_w\gamma_w$ (Table 4). The difference in OA enhancement
ratios for these varying $C_w/M_w\gamma_w$ is as much as 119% if estimated using the upper-
bound chemistry assumptions (Fig. 4a) and as much as 63% for the lower-bound
assumptions (Fig. 4b). For the upper-bound-chemistry simulations, OAER$_{inert}$ for the
simulations using $C_w/M_w\gamma_w$ of 20 and 9 µmole m$^{-3}$ and Krechmer's values (1.6, 1.9 and
1.9) are close to the experimental values (1.7±0.7) reported by Hennigan et al. (2011),
suggesting our simulations using these parameter settings could reflect the conditions in
the chamber experiments. Generally, the lower-bound-chemistry simulations all
underpredict the experimental range of Hennigan et al. (2011). Most of those
simulations result in a net loss of OA (OAER$_{inert}$ less than 1), although the simulations
with the Zhang $C_w/M_w\gamma_w$ set overlap with the low end of the Hennigan et al. (2011)
range.

The vapor accommodation coefficient with the walls, $\alpha_w$, has also been demonstrated to
be an important parameter in chambers that influences the vapor-wall loss rates (Zhang
et al. 2014; Bian et al., 2015). A value of 1 represents perfect accommodation,
representing no limitation on the vapor-wall loss rates due to this process. Based on
their series of lab studies, Matsunaga and Ziemann (2010) recommended values of $\alpha_w$
larger than 10$^{-5}$. Zhang et al. (2014) and Bian et al. (2015) both showed the insensitivity
of vapor wall loss to $\alpha_w$ when $\alpha_w > 10^{-4}$, but vapor wall loss was largely suppressed
using the varying $\alpha_w$ as a function of $C^*$ that was suggested by Zhang et al. (2015). We
thus simulate the experiments for choices of $\alpha_w = 1$ and for varying $\alpha_w$, as sensitivity
tests from our previously assumed value of 10$^{-5}$. $C_w/M_w\gamma_w$ is set to Krechmer's values for
this series of simulations. Fig. 5 shows that assuming $\alpha_w = 1$ decreases OAER$_{inert}$ by 18-
31% compared with the base-case simulations using $\alpha_w = 1×10^{-5}$, since $k_{on}$ is nearly one





order of magnitude higher for $\alpha_w = 1$ than for $\alpha_w = 1\times10^{-5}$ (Table 3). On the other hand,
         OAER$_{inert}$ nearly doubles when using the varying $\alpha_w$ relative to the $1\times10^{-5}$ simulations,
as vapor wall loss is slower on average for the varying $\alpha_w$ (i.e. $3.7\times10^{-9}$ to $1.1\times10^{-6}$ for
         our simulated $C^*$ range). Compared with the experimental values of Hennigan et al.
(2011), it appears that using $\alpha_w$ of $1\times10^{-5}$, or the varying $\alpha_w$ values with the lower-bound-
         chemistry assumptions, can better represent the FLAME-III experiments; however, we
are unable to determine which set of $\alpha_w$, $C_w/M_w\gamma_w$, and chemistry assumptions best
         represent the actual processes occurring in the chamber, since different combinations
of these values can reproduce the observed OAER$_{inert}$ range.

         Whether the upper- or lower-bound chemical mechanism assumptions are applied, our
simulations show that OAER$_{inert}$ increases significantly for most of the cases when vapor
         wall losses are shut off, implying that vapor-wall-loss suppression of SOA formation is a
robust result across our simulations (Fig. 4). For example, OAER$_{inert}$ for the upper-
         bound-chemistry simulations without vapor wall loss is $3.3\pm0.7$ (Fig. 4a), or over a 200%
increase in OA attributable to chemical formation of SOA from species that are lost to
         the walls in typical experiments. Most of the measurements and simulations including
vapor wall losses result in OA increases due to SOA formation of 100% or less. Thus,
         our simulations imply that SOA production in biomass-burning-smoke SOA laboratory
smog chamber experiments may be underestimated by a factor of 2 or more due to
         vapor wall losses, and that applying lab-derived apparent SOA formation rates to
simulations of the evolution of ambient OA would similarly underestimate the impacts of
         photo-oxidation of biomass-burning products. We explore these potential atmospheric
impacts in the next section.

         3.3 SOA production in ambient plumes

The semi-volatile nature of organics from biomass burning not only complicates SOA
         estimation from chamber studies, but also can influence OA evolution during plume
transport and dilution. In dispersion, the initial plume cross-sectional area is a key factor
         that determines the relative plume dilution rate during transport (Sakamoto et al., 2016).
The initial plume width is associated with fire size, which means that the fire size could
         largely influence the plume evolution (Sakamoto et al., 2016). We perform simulations
on the evolution of ambient OA concentrations over 4 hours of simulated transport, for
         four different fire areas of $1\times10^{-4}$, $1\times10^{-2}$, $1\times10^{0}$ and $1\times10^{2}$ km$^2$ (with the fire width
assumed to be the square root of these areas). In these simulations, we set the mass
         flux to $5\times10^{-6}$ kg m$^{-2}$ s$^{-1}$ and the atmospheric stability to the neutral atmospheric Pasquill
stability condition, $D$. The initial mass concentrations for different-sized fires are
         assumed to be similar in all cases (~$10^3$ µg m$^{-3}$). The simulated time evolution of various
key quantities is shown for each of the four different fire sizes in Fig. 6, with the upper-
         bound chemistry cases shown as solid lines and the lower-bound chemistry as dotted
lines.



The organic mass (OM) concentration in the gas and particle phases predicted for the small fire ($1\times10^{-4}$ km$^2$, prescribed fire size) drops quickly from $1\times10^3$ to $3\times10^{-3}$ µg m$^{-3}$ over the four simulated hours (blue lines, Figs. 6a and b) due to the strong dilution: a dilution ratio of over $10^5$ with respect to the initial volume is achieved within 2 hours, as shown in Fig 6c. The OA concentration for the large fire ($1\times10^2$ km$^2$, wildfire size) decreases from around $3\times10^3$ to $1\times10^3$ µg m$^{-3}$ because of weak dilution (dilution ratio <10). OAER$_{inert}$ increases to around 1.06-1.20 (depending on upper- versus lower-bound chemistry) for the 100 km$^2$ fire area; however, for the smaller fires, OAER$_{inert}$ initially decreases due to the dominant role of OA evaporation driven by dilution, but eventually recovers as SOA formation rates exceed the loss rates (particularly for the upper-bound-chemistry simulations, Fig. 6d). The upper-bound-chemistry simulated OAER$_{inert}$ after 4 h transport are all above 1, while OAER$_{inert}$ remains below 1 for the small fires in the lower-bound-chemistry simulations. Thus, the range in the simulations shown in Fig. 6d captures the range in the competition between OA evaporation due to dilution and OA formation due to chemistry and condensation. Interestingly, OAER$_{inert}$ evolves virtually identically for the two smallest fires (Fig. 6d), despite different dilution ratios (Fig. 6c) due to the biomass-burning OA concentrations dropping below the concentration background OA entrained into the plume (5 µg m$^{-3}$) in both plumes, which suggests that the background OA concentration also plays a role affecting the OAER values.

Atmospheric stability is an important parameter that influences the dilution rate. Figs 7 and 8 show the impacts on the predictions of changing atmospheric stability for low (Fig. 7) and high (Fig. 8) emission mass fluxes ($2\times10^{-8}$ and $5\times10^{-6}$ kg m$^{-2}$s$^{-1}$), all for moderate 1 km$^2$ fire areas. Unstable atmospheres (stability-class A) favor the vertical and horizontal mixing of air parcels that enhances dilution (Fig 7c). Stable atmospheres (stability-class F) resist vertical mixing and have weaker dilution. Therefore, OA evolution in unstable atmospheres (A) behaves qualitatively similar to the small fires in Fig. 7 and has a similar decreasing-then-increasing pattern for OAER$_{inert}$. OA evolution in stable atmospheres (F) behaves qualitatively similar to the large fires in Fig. 7, leading to a steady increase in OAER$_{inert}$ with time (Fig. 7). For the low-emission mass flux (Fig. 8), OAER$_{inert}$ shows a similar pattern across all stability classes, increasing steadily with time. This monotonic increase arises because the plumes begin in a dilute state where the biomass-burning OA concentrations quickly drop below the background non-volatile BC concentrations entrained into the plume (5 µg m$^{-3}$). In this limit, further dilution does not lead to further evaporation, so in each of the stability cases chemistry exceeds evaporation. Again, this shows that the results should be sensitive to background non-volatile OA concentrations.

The sensitivity tests shown in Figs. 6-8 demonstrate that OA enhancement ratios measured in the field using BC or CO as a conserved tracer (OAER$_{inert}$) may undergo





very different trajectories based on (1) the fire size, (2) the emissions mass flux, and (3) the stability of the atmosphere - even when the OA volatility distribution and chemical
mechanisms are identical. This variance with fire size, mass flux, and stability may explain at least some of the variability in the measured time evolution of OA
enhancement ratios ($OAER_{inert}$) reported in field studies.

### 3.4. Is the traditional OA enhancement ratio reported in field studies a good proxy for
SOA formation? $OAER_{inert}$ versus $OAER_{chem}$

As described earlier, $OAER_{inert}$ (the OA enhancement ratio calculated by using an inert
tracer, such as BC, to account for physical dilution) and $OAER_{chem}$ (the OA enhancement ratio calculated comparing simulations with chemistry on versus
chemistry off) differed for our simulations of smog-chamber experiments with vapor wall losses on. We find that the differences between $OAER_{chem}$ and $OAER_{inert}$ can be even
more dramatic in our plume simulations. Fig. 6 shows that $OAER_{chem}$ increases steadily across all four different-sized fires. Unlike $OAER_{inert}$, which had the largest increases for
the large fire, $OAER_{chem}$ has the largest increases for small fires, reaching values of 2.2 for the small fires and 1.3 for the large fires (with upper-bound chemistry). More organic
material is evaporated from particles in plumes of smaller fires, which gives a larger reservoir of SOA precursors to generate SOA, compared to the plumes of larger fires.
Thus, while $OAER_{inert}$ estimates that are traditionally reported in field studies may show values similar to or less than 1, the OA in these plumes may actually be strongly
enhanced by SOA formation, and indeed evaporation of precursors driven by dilution is required to replenish the reservoir of SOA precursors in the gas phase so that these
processes are not only in competition but are dependent on each other. In cases where little apparent SOA production is occurring, our studies suggest that SOA formation is
simply balancing the loss of OA from evaporation. This explanation is consistent with the findings from some observational studies reporting increased oxygenation with time
for the OA in sampled biomass burning plumes, but lower average $\Delta OA / \Delta CO$ (or a decreasing $OAER_{inert}$) in aged plumes (Jolleys et al., 2015).

Analogous results are shown for the influence of atmospheric stability in Fig. 7. The $OAER_{inert}$ values are largest for the most-stable conditions. On the other hand
$OAER_{chem}$ values are largest for the least-stable conditions that have the most organic-vapor evaporation generating the largest pool of SOA precursor vapors. Under low
emission-flux conditions (Fig. 8), the plume is already dilute upon emission and thus both $OAER_{inert}$ and $OAER_{chem}$ have nearly identical values, monotonically increasing
with transport time.

This comparison of $OAER_{inert}$ and $OAER_{chem}$ shows that $OAER_{inert}$ computed from field
measurements may not be indicative of the relative amount of SOA formed in the plume, due to competition with OA loss to dilution. Further, the relationship between $OAER_{inert}$



and OAER$_{chem}$ can depend greatly on the fire size, smoke emission flux, and the atmospheric stability, and different conclusions regarding the efficiency and impact of
photooxidation can be drawn for the same fuels, combustion phases, and chemical mechanisms if the emissions are sampled under those varying fire size and
environmental conditions.

4. Summary and Conclusions

We investigated the processes controlling biomass-burning OA evolution in smog
chambers and in ambient plumes. We used aerosol microphysics simulations with resolved organic volatility, kinetic condensation/evaporation, and gas-phase chemistry
(ignoring potential particle- and heterogeneous-phase chemistry) to explore these processes. We found that differences seen between laboratory and field observations
may be explained, in part, due to processes that control OA evaporation (and SOA precursor losses) in these experiments.

For laboratory smog-chamber experiments in Teflon chambers (specifically the FLAME-III experiments reported by Hennigan et al., 2011), our simulations showed that
vapor wall losses remove SOA precursor vapors and drive OA evaporation. Uncertainties in parameters that control vapor wall losses, such as the wall saturation
concentration and wall accommodation coefficient, as well as uncertainties in gas-phase chemistry, lead to uncertainties in our simulations. We are able to reproduce the
observed OA concentration profiles from the FLAME-III experiments using a range of wall-loss and chemistry parameters that fall within previously published estimates, but
there is no unique set of parameters that can be identified at this time. However, under all assumed parameters, the apparent SOA formation was suppressed by vapor wall
losses. For the simulations that best reproduced the OA concentration profiles from the FLAME-III experiments, we found that turning off vapor wall losses in these simulations
leads to 2-3x increases in the total apparent SOA production in the experiment. Thus, vapor-phase wall losses should be considered and corrected for in biomass-burning
SOA smog-chamber experiments.

    For ambient expanding plumes, we showed through similar simulations with
594 identical gas-phase chemistry assumptions that the fire area, mass emissions flux, and atmospheric stability strongly modulate initial plume concentrations and plume dilution
rates. Conditions with fast dilution (small fire areas and unstable atmospheric conditions) drive faster OA evaporation relative to slow-dilution conditions. However, the
evaporated OA serves as precursor vapors for SOA formation. Thus, quickly diluting plumes may have substantial initial drops in the ratio of OA to inert tracers (relative to



slowly diluting plumes), but the ratio of OA to inert tracers later increases more rapidly in the quickly diluting plumes due to the faster SOA formation.

To decouple the influences of POA evaporation and SOA formation on the evolution of the net OA, we defined two metrics: (1) $OAER_{inert}$, which uses an inert

tracer (e.g. CO or BC) to normalize OA in the plume, as is commonly done in laboratory and field experiments, and (2) $OAER_{chem}$, which uses a simulation with chemistry turned

off to normalize the OA in the plume, which is generally only possible in modeling studies. While $OAER_{inert}$ is influenced by both POA evaporation and SOA condensation,

$OAER_{chem}$ shows influence of SOA condensation which allowed us to decouple the influence of POA evaporation and SOA condensation. Through these two metrics, we

showed that many plumes with $OAER_{inert}$ values near 1 (implying little net change in OA) may be strongly influenced by SOA production that is balanced by POA evaporation.

We found the SOA-production influence to be strongest for rapidly diluting plumes (such as those from small-area fires or under unstable atmospheric conditions), where SOA

may contribute to a doubling of OA concentrations within 4 hours relative to a simulation with chemistry off, even though field measurements might have observed little to no net

change in OA in the plume with time.

Our results highlight that the evolution of OA in the atmosphere depends on more

than the details of the fuel types and the combustion efficiency of those fuels, yet these fuel/combustion details are often the focus of many experiments. The size of the fire

and the meteorological conditions may also influence whether a net OA increase or decrease is inferred, when dilution alone is accounted for by normalizing with inert-

tracer concentrations. The large range in reported observed OA changes in experiments and ambient plume profiles (e.g., Grieshop et al., 2009; Yokelson et al., 2009; Cubison

et al., 2011;Hennigan et al. 2011; Akagi et al., 2012;Ortega et al., 2013; May et al., 2015) may be explained, in part, by these factors. Additionally, as we used identical chemistry

assumptions in all of our simulations, we showed that the changes in OA with time in laboratory and field experiments cannot easily be compared to each other due to

different influences of chamber walls and plume dilution. The apparent observed OA evolution in the laboratory and field may be drastically different (e.g. showing a net gain

in the lab while showing a net loss in the field) even with identical chemical mechanisms and rates in the laboratory and field experiments. These findings may also explain in

part the systematic inconsistencies in reported OA enhancements measured in the laboratory and in field experiments (e.g., Jolleys et al., 2014). Thus, laboratory and field

observations require a thorough understanding of the processes that drive OA evaporation (and SOA-precursor losses) before the impact of photochemical SOA

production can be isolated and quantified.



*Acknowledgement*

This study was supported by the Joint Fire Science Program (JFSP) under projects of 14-1-03-26 and 14-
1-03-44. We thank the Fire Lab At Missoula Experiment (FLAME) III project team and Cyle Wold, Emily
Lincoln, and Wei Min Hao from the FSL for their support in organizing and conducting the FLAME-III
study and for providing the data set used here. We thank Chris Hennigan, Gabriella Engelhart, Marissa
Miracolo, Albert Presto, and Allen Robinson for their smog-chamber measurements during FLAME-III.
Funding for the FLAME-III project was provided by the National Park Service, JFSP and the EPA STAR
program through the National Center for Environmental Research (NCER) under grants R833747 and
R834554, and DOE (BER, ASR program) DE-SC0006035. The views, opinions, and/or findings contained
in this paper are those of the authors and should not be construed as an official position of the funding
agencies. Although the research described in this article has been funded in part by the United States
Environmental Protection Agency, it has not been subjected to the Agency's required peer and policy
review and therefore does not necessarily reflect the views of the Agency and no official endorsement
should be inferred.

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

822





Table 1. Data for 18 wood smoke samples introduced to the smog chamber, including fuel types, initial number concentration and corresponding size distribution parameters (median diameter in nm and geometric standard deviation, σ), initial total aerosol nonrefractory mass concentration, the organic mass fraction of the aerosol phase and OH exposure rate. The Burn ID and OH exposure refer to the schedule of burns in FLAME III, as reported in Hennigan et al. (2011).

| Burn ID | Fuel type | Temp (K) | Initial particle number concentration (cm$^{-3}$) | Num. size dist. Median diameter (nm) | σ | Initial total mass concentration[1] (μg m$^{-3}$) | Organic mass fraction[2] | $k_{w,p0}$ (s$^{-1}$) | $k_e$ (s$^{-1}$) | OH exposure (molecules cm$^{-3}$ s) |
|---|---|---|---|---|---|---|---|---|---|---|
| 37 | Lodgepole Pine | 292.9 | 5843 | 157 | 1.73 | 44.96 | 0.943 | 8.03×10$^{-5}$ | 1.07 | 1.56×10$^{10}$ |
| 38 | Lodgepole Pine | 286.8 | 7612 | 127 | 1.67 | 40.96 | 0.896 | 6.27×10$^{-5}$ | 1.41 | 1.40×10$^{10}$ |
| 40 | Ponderosa Pine | 279.5 | 6505 | 160 | 1.84 | 63.73 | 0.954 | 8.67×10$^{-5}$ | 0.69 | 2.71×10$^{10}$ |
| 42 | Wire Grass | 277.0 | 8107 | 123 | 1.55 | 19.63 | 0.484 | 1.07×10$^{-4}$ | 0.77 | 3.50×10$^{10}$ |
| 43 | Saw Grass | 284.2 | 5406 | 123 | 1.73 | 18.16 | 0.347 | 1.07×10$^{-4}$ | 0.52 | 3.10×10$^{10}$ |
| 45 | Turkey Oak | 286.3 | 6334 | 106 | 1.63 | 16.80 | 0.506 | 8.11×10$^{-5}$ | 0.99 | 2.09×10$^{10}$ |
| 47 | Gallberry | 286.7 | 8265 | 123 | 1.61 | 39.16 | 0.881 | 7.37×10$^{-5}$ | 0.19 | 6.12×10$^{10}$ |
| 49 | Sage | 285.0 | 5486 | 127 | 1.71 | 17.76 | 0.321 | 8.84×10$^{-5}$ | 0.84 | 1.84×10$^{10}$ |
| 51 | Alaskan Duff | 282.5 | 4175 | 88 | 1.83 | 20.38 | 0.898 | 7.00×10$^{-5}$ | 0.32 | [3]4.29×10$^{10}$ |
| 53 | Sage | 287.2 | 5619 | 132 | 1.76 | 16.09 | 0.348 | 8.43×10$^{-5}$ | 0.91 | [3]4.29×10$^{10}$ |
| 55 | White Spruce | 281.6 | 4641 | 115 | 1.83 | 27.73 | 0.761 | 8.13×10$^{-5}$ | 0.31 | 6.59×10$^{10}$ |
| 57 | Ponderosa Pine | 277.9 | 6624 | 161 | 1.81 | 72.83 | 0.935 | 8.43×10$^{-5}$ | 0.96 | 7.99×10$^{10}$ |
| 59 | Chamise | 281.9 | 7173 | 148 | 1.79 | 24.89 | 0.221 | 7.58×10$^{-5}$ | 0.83 | 4.95×10$^{10}$ |
| 61 | Lodgepole Pine | 283.1 | 6059 | 153 | 1.79 | 63.03 | 0.944 | 6.30×10$^{-5}$ | 0.29 | 7.89×10$^{10}$ |
| 63 | Pocosin | 277.9 | 7463 | 112 | 1.65 | 26.20 | 0.603 | 8.46×10$^{-5}$ | 0.37 | 8.22×10$^{10}$ |
| 65 | Gallberry | 275.3 | 7763 | 159 | 1.68 | 85.98 | 0.899 | 1.43×10$^{-4}$ | 0.62 | 4.94×10$^{10}$ |
| 66 | Black Spruce | 279.0 | 9828 | 96 | 1.66 | 35.21 | 0.852 | 1.02×10$^{-4}$ | 0.36 | 2.63×10$^{10}$ |
| 67 | Wire Grass | 274.5 | 11580 | 129 | 1.52 | 36.51 | 0.619 | 5.78×10$^{-5}$ | 0.28 | 3.06×10$^{10}$ |



[1]total mass = [OA] + [SO$_4^{2-}$] + [NO$_3^-$] + [NH$_4^+$] + [Cl$^-$] + [BC], total aerosol non-refractory mass concentration as measured by the Aerodyne quadruple aerosol mass spectrometer and black carbon was determined by a seven-channel Aethalometer at 880 nm.
[2]organic fraction = [OA] / ([OA] + [SO$_4^{2-}$] + [NO$_3^-$] + [NH$_4^+$] + [Cl$^-$] + [BC])
[3]We have assumed the average OH exposure of the other 16 experiments, as no OH exposure rate was provided for these two experiments.





Table 2. Gas-phase chemistry volatility matrix that describes the change in volatility of the gas-phase organics after a single reaction with OH. Labels a and b represent the cases with four- and two-volatility-bin drops per reaction, respectively.

| Precursor $\log_{10}C^*$ ($\mu g\ m^{-3}$) | Product $\log_{10}C^*$ ($\mu g\ m^{-3}$) | | | | | | | | | | | | |
|---|---|---|---|---|---|---|---|---|---|---|---|---|
| | -3 | -2 | -1 | 0 | 1 | 2 | 3 | 4 | 5 | 6 | 7 | 8 | 9 |
| -2 | a, b | | | | | | | | | | | | |
| -1 | a, b | | | | | | | | | | | | |
| 0 | a | b | | | | | | | | | | | |
| 1 | a | | b | | | | | | | | | | |
| 2 | | a | | b | | | | | | | | | |
| 3 | | | a | | b | | | | | | | | |
| 4 | | | | a | | b | | | | | | | |
| 5 | | | | | a | | b | | | | | | |
| 6 | | | | | | a | | b | | | | | |
| 7 | | | | | | | a | | b | | | | |
| 8 | | | | | | | | a | | b | | | |
| 9 | | | | | | | | | a | | b | | |
| 10 | | | | | | | | | | a | | b | |
| 11 | | | | | | | | | | | a | | b |





Table 3. Input parameters for the ambient-plume Gaussian dispersion simulations.

| Parameter | Description | Value |
|---|---|---|
| $D_p$ | Emission particle dry diameter, μm | 0.157 |
| σ | Emission particle size distribution standard deviation | 1.7 |
| $k_{OH}$ | Ambient reaction rate constant, $cm^3\ molecule^{-1}\ s^{-1}$ | upper: $-5.70 \times 10^{-12} \ln(C^*) + 1.14 \times 10^{-10}$<br>lower: $-1.84 \times 10^{-12} \ln(C^*) + 4.27 \times 10^{-10}$ |
| [OH] | Ambient OH concentration, molecules $cm^{-3}$ | $1.08 \times 10^6$ |
| Mass Flux | Emission mass flux from fire, $kg\ m^{-2}s^{-1}$ | $2 \times 10^{-8}$, $5 \times 10^{-6}$ |
| Fire area | Fire emissions area, $km^2$ | $1 \times 10^2$, 1, $1 \times 10^{-2}$, $1 \times 10^{-4}$ |
| Wind speed | Mean boundary-layer wind speed, $ms^{-1}$ | 5 |
| Stability class | Pasquill stability classes for atmospheric turbulence | A, D, F |
| Boundary height | Mean boundary height, m | 2500 |
| T | Ambient temperature during dilution, K | 298 |
| $Mass_{bg}$ | Background aerosol mass concentration, μg $m^{-3}$ | 5.0 |
| $D_{p,bg}$ | Dry diameter of background particles, μm | 0.3 |
| $σ_{p,bg}$ | Geometric standard deviation of size distribution of background particles, μm | 1.8 |





Table 4. Vapor wall-loss rate constants (s$^{-1}$, $k_{w,on}$ and $k_{w,\,off}$) for each volatility bin for cases with varying $C_w/M_w\gamma_w$ (Krechmer et al., 2016), for different XX $\alpha_w$ as shown; last column is for the case varying $C_w/M_w\gamma_w$ as in Zhang et al., (2015).

| log$_{10}$C* | varying $C_w/M_w\gamma_w$ (Krechmer et al., 2016); $\alpha_w=1\times10^{-5}$ | | varying $C_w/M_w\gamma_w$ (Krechmer et al., 2016); $\alpha_w=1$ | | varying $C_w/M_w\gamma_w$ (Krechmer et al., 2016); varying $\alpha_w$ (Zhang et al. 2015) | | varying $C_w/M_w\gamma_w$ (Zhang et al., 2015); $\alpha_w=1\times10^{-5}$ | |
|---|---|---|---|---|---|---|---|---|
| | $k_{on}$ | $k_{off}$ | $k_{on}$ | $k_{off}$ | $k_{on}$ | $k_{off}$ | $k_{on}$ | $k_{off}$ |
| -3 | $7.33\times10^{-4}$ | $2.01\times10^{-8}$ | $4.01\times10^{-3}$ | $1.10\times10^{-7}$ | $1.55\times10^{-4}$ | $4.26\times10^{-9}$ | $7.33\times10^{-4}$ | $9.90\times10^{-5}$ |
| -2 | $7.58\times10^{-4}$ | $2.26\times10^{-7}$ | $4.02\times10^{-3}$ | $1.20\times10^{-6}$ | $1.05\times10^{-4}$ | $3.15\times10^{-8}$ | $7.58\times10^{-4}$ | $1.58\times10^{-4}$ |
| -1 | $7.86\times10^{-4}$ | $2.56\times10^{-6}$ | $4.02\times10^{-3}$ | $1.31\times10^{-5}$ | $7.15\times10^{-5}$ | $2.33\times10^{-7}$ | $7.86\times10^{-4}$ | $2.55\times10^{-4}$ |
| 0 | $8.18\times10^{-4}$ | $2.94\times10^{-5}$ | $4.03\times10^{-3}$ | $1.45\times10^{-4}$ | $4.86\times10^{-5}$ | $1.75\times10^{-6}$ | $8.18\times10^{-4}$ | $4.16\times10^{-4}$ |
| 1 | $8.54\times10^{-4}$ | $8.61\times10^{-5}$ | $4.03\times10^{-3}$ | $4.07\times10^{-4}$ | $3.31\times10^{-5}$ | $3.34\times10^{-6}$ | $8.54\times10^{-4}$ | $6.91\times10^{-4}$ |
| 2 | $8.97\times10^{-4}$ | $2.57\times10^{-4}$ | $4.04\times10^{-3}$ | $1.16\times10^{-3}$ | $2.27\times10^{-5}$ | $6.51\times10^{-6}$ | $8.97\times10^{-4}$ | $1.16\times10^{-3}$ |
| 3 | $9.47\times10^{-4}$ | $7.85\times10^{-4}$ | $4.06\times10^{-3}$ | $3.36\times10^{-3}$ | $1.56\times10^{-5}$ | $1.30\times10^{-5}$ | $9.47\times10^{-4}$ | $2.02\times10^{-3}$ |
| 4 | $1.01\times10^{-3}$ | $2.47\times10^{-3}$ | $4.07\times10^{-3}$ | $9.97\times10^{-3}$ | $1.09\times10^{-5}$ | $2.68\times10^{-5}$ | $1.01\times10^{-3}$ | $3.58\times10^{-3}$ |
| 5 | $1.09\times10^{-3}$ | $6.50\times10^{-3}$ | $4.09\times10^{-3}$ | $2.45\times10^{-2}$ | $7.75\times10^{-6}$ | $4.63\times10^{-5}$ | $1.09\times10^{-3}$ | $6.68\times10^{-3}$ |
| 6 | $1.10\times10^{-3}$ | $6.90\times10^{-2}$ | $4.10\times10^{-3}$ | $2.56\times10^{-1}$ | $5.10\times10^{-6}$ | $3.19\times10^{-4}$ | $1.10\times10^{-3}$ | $1.01\times10^{-2}$ |
| 7 | $1.10\times10^{-3}$ | $6.90\times10^{-1}$ | $4.10\times10^{-3}$ | $2.56\times10^{0}$ | $3.28\times10^{-6}$ | $2.05\times10^{-3}$ | $1.10\times10^{-3}$ | $1.43\times10^{-2}$ |
| 8 | $1.10\times10^{-3}$ | $6.90\times10^{0}$ | $4.10\times10^{-3}$ | $2.56\times10^{1}$ | $2.12\times10^{-6}$ | $1.32\times10^{-2}$ | $1.10\times10^{-3}$ | $2.05\times10^{-2}$ |
| 9 | $1.10\times10^{-3}$ | $6.90\times10^{1}$ | $4.10\times10^{-3}$ | $2.56\times10^{2}$ | $1.36\times10^{-6}$ | $8.47\times10^{-2}$ | $1.10\times10^{-3}$ | $2.91\times10^{-2}$ |
| 10 | $1.10\times10^{-3}$ | $6.90\times10^{2}$ | $4.10\times10^{-3}$ | $2.56\times10^{3}$ | $8.72\times10^{-7}$ | $5.45\times10^{-1}$ | $1.10\times10^{-3}$ | $4.12\times10^{-2}$ |
| 11 | $1.10\times10^{-3}$ | $6.90\times10^{3}$ | $4.10\times10^{-3}$ | $2.56\times10^{4}$ | $5.61\times10^{-7}$ | $3.50\times10^{0}$ | $1.10\times10^{-3}$ | $5.87\times10^{-2}$ |





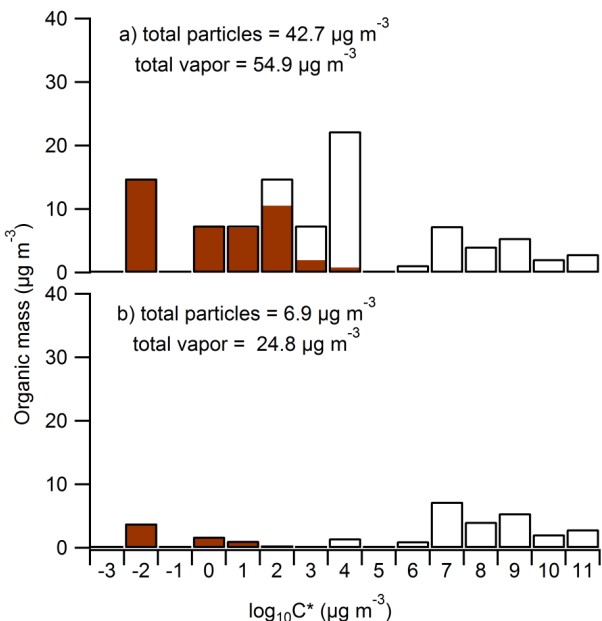

Figure 1. a) Volatility distribution with 15 volatility bins, adapted from the work of May et al. (2013) and Hatch et al. (2016). The average total initial organic aerosol mass concentration is 42.7 µg m$^{-3}$ over the 18 experiments. For this mass concentration, the shaded area represents the organic mass in the particulate phase in each volatility bin. b) The simulated volatility distribution without chemistry after 4 hr of particle and vapor wall loss. The concentrations are the means across all 18 experiments.





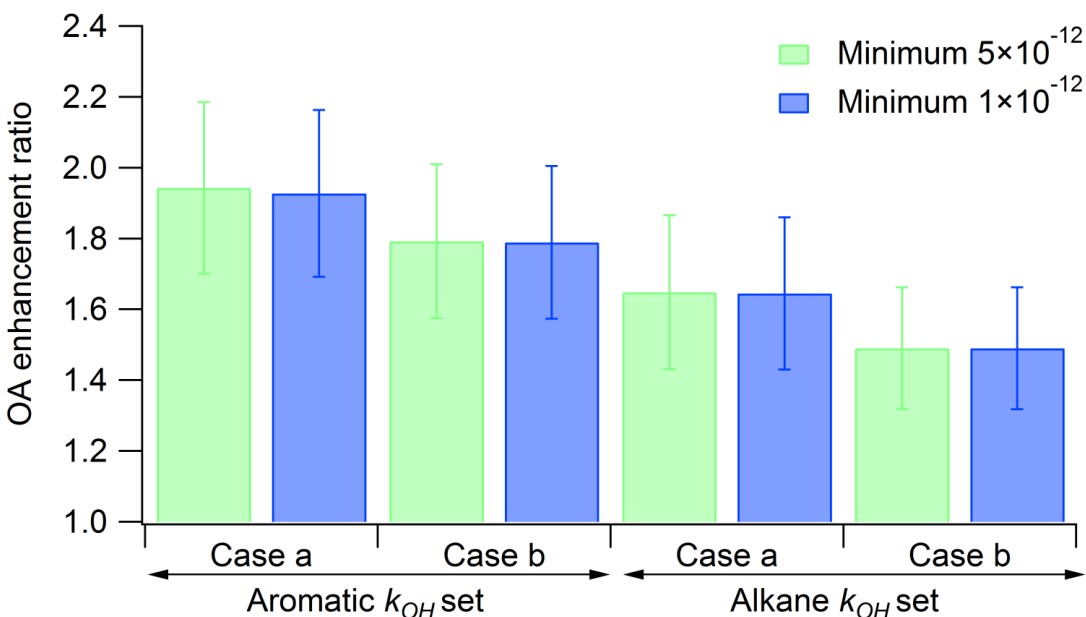

Figure 2. OA enhancement ratios (OAER$_{inert}$ and OAER$_{chem}$ are equivalent in these simulations), in the absence of particle and vapor wall losses, averaged over the 18 experimental simulations using $k_{OH}$ sets fitted for aromatics and alkanes with a four-volatility-bin drop per reaction (Case a) and a two-volatility bin drop per reaction (Case b). The minimum k$_{OH}$ value is set to be $5\times10^{-12}$ (green bars) and $1\times10^{-12}$ (blue bars) cm$^3$ mole$^{-1}$ s$^{-1}$, respectively. The error bars represent one standard deviation across the 18 simulations and represent experiment-to-experiment variability.





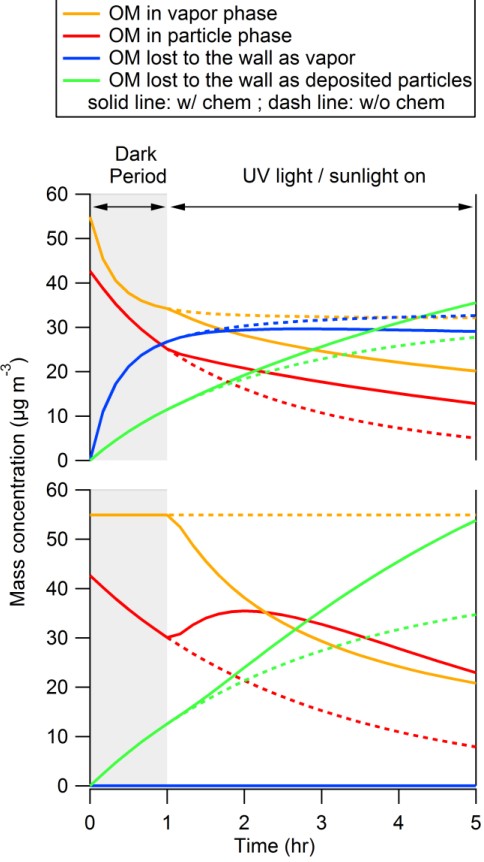

Figure 3. Time evolution of organic mass (OM, in units of $\mu g \ m^{-3}$) in the vapor phase (gold lines) and particulate phase (red lines), averaged over the 18 simulations, assuming no chemical reactions occurring (dashed lines) and including oxidation reactions (solid lines). Simulations with chemistry on use $k_{OH}$ fitted for aromatics with a four-volatility-bin- drop in volatility assumed for the products. a) with particle and vapor wall loss on; b) with vapor wall loss off. Particle-phase wall losses are included in both simulations; the masses of particles and vapors lost to the walls have been normalized by the volume of the bag to obtain mass concentration units. The simulations use Krechmer's saturation concentrations ($C_w/M_w\gamma_w$) (Krechmer et al. 2016) and a mass accommodation coefficient of $1\times10^{-5}$. In all cases, the first hour simulates the process of primary organic aerosol characterization in the dark (no chemical reactions).


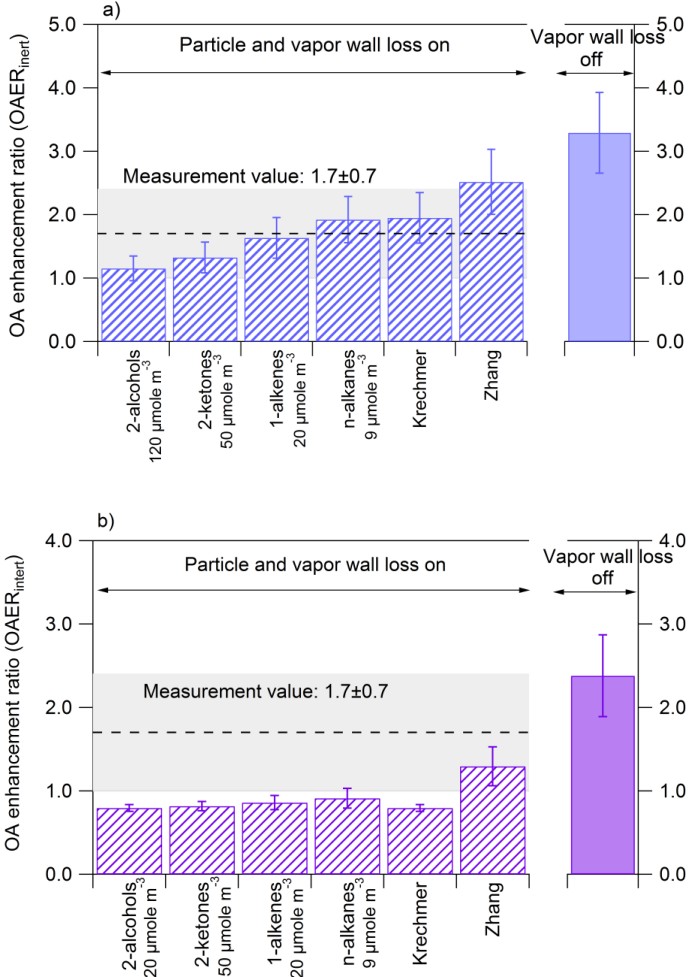

Figure 4. OAER$_{inert}$ enhancement ratios in the simulations, as calculated from Eqn 7, using saturation concentrations ($C_w/M_w\gamma_w$) of 120, 50, 20, and 9 µmole m$^{-3}$ as suggested by Matsunaga and Ziemann (2010), and for varying $C_w/M_w\gamma_w$ as suggested by Krechmer et al. (2016) and Zhang et al. (2015). Two sets of reaction rates have been applied: a) upper-bound chemistry ($k_{OH}$ set for aromatics with four-volatility-bin drop per reaction) and b) lower-bound chemistry ($k_{OH}$ set for alkanes with two-volatility-bin drop per reaction). The mass accommodation coefficient is set to 1×10$^{-5}$ in all simulations. The striped bars represent the simulations with particle and vapor wall loss on and the solid bars represent the simulations with vapor wall loss off. The dashed line and grey area represent the measurement value and its standard deviation from Hennigan et al. (2011).





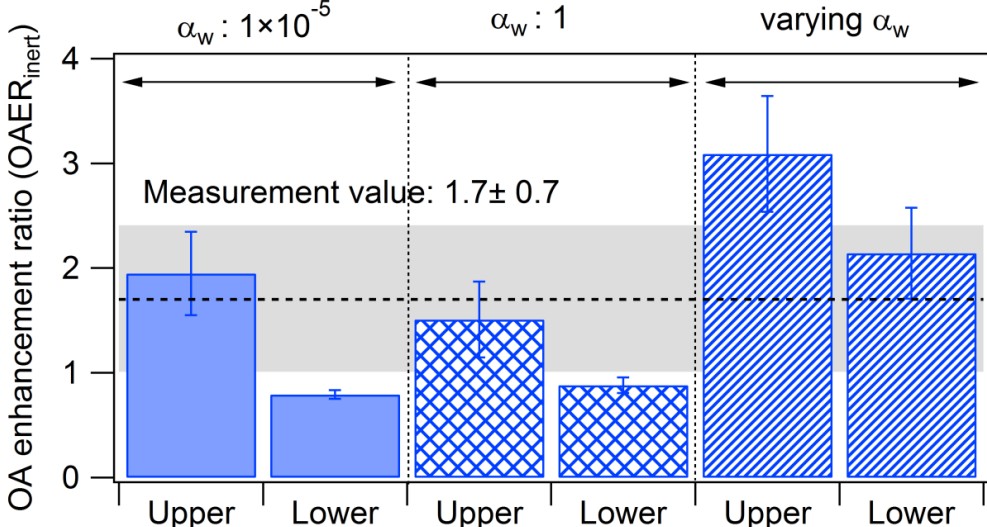

Figure 5. The effect of variable mass accommodation coefficients on the OAER$_{inert}$ enhancement ratios shown in Fig. 4. All simulations used varying $C_w/M_w\gamma_w$ (Krechmer et al. 2016). Results for upper- and lower-bound chemistry assumptions are shown, with assumed $\alpha_w$ of $1\times10^{-5}$ (solid bars), 1 (gridded bars) and varying $\alpha_w$ as a function of $C^*$ (striped bars, Zhang et al., 2015). The dashed line and grey area represent the measurement value and its standard deviation from Hennigan et al. (2011).



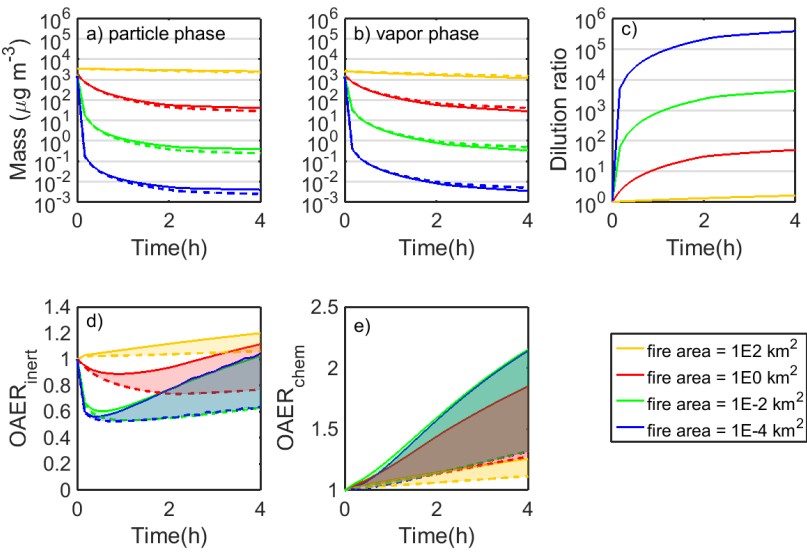

Figure 6. Time evolution of a) organic mass (OM) in the particle phase, b) OM in the vapor phases, c) dilution ratios, d) $OAER_{inert}$ and e) $OAER_{chem}$ during Gaussian dispersion, using the parameters listed in Table 3 with fire areas of 100, 1, $1\times10^{-2}$ and $1\times10^{-4}$ km$^2$ and an emission flux of $5\times10^{-6}$ kg m$^{-2}$ s$^{-1}$. Solid lines represent the upper-bound-chemistry simulations and dashed lines represent the lower-bound-chemistry simulations. Shaded areas bound the ranges of estimated OA enhancement.





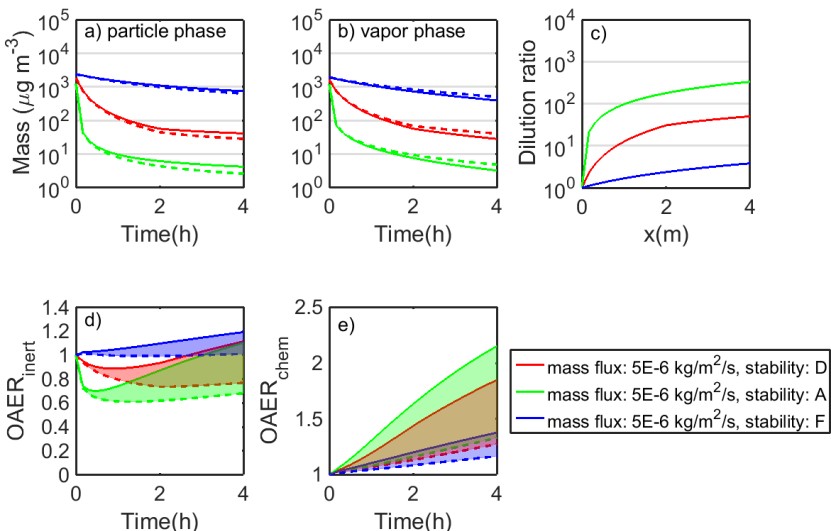

Figure 7. Time evolution during Gaussian dispersion of a) organic mass (OM) in the particle phase, b) OM in the vapor phases, c) dilution ratio, d) $OAER_{inert}$ , and e) $OAER_{chem}$, with a fire area of 1 km$^2$, a mass flux (ML) of $5 \times 10^{-6}$ kg m$^{-2}$ s$^{-1}$, and assuming different atmospheric stability classes (A, D, and F; see Table 3).





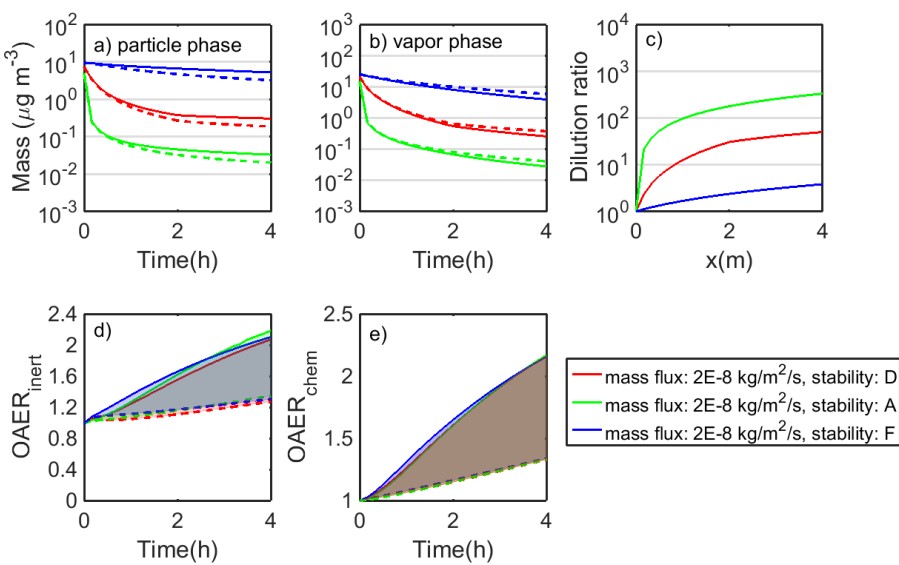

Figure 8. As in Fig. 7, but for an assumed mass flux of 2×10⁻⁸ kg m⁻² s⁻¹.