# Peer review of "Secondary organic aerosol formation in biomass-burning plumes: Theoretical analysis of lab studies and ambient plumes"

_Atmospheric Chemistry and Physics, 2016_

## Referee Comment (RC1) · Anonymous Referee #1 · 2 Dec 2016

Review of "Secondary organic aerosol formation in biomass-burning plumes: Theoretical analysis of lab studies and ambient plumes" by Bian et al.

**General Comments**
This paper presents a modeling analysis of SOA formation in aging biomass burning (BB) plumes. The study presents several new angles on this topic, namely a) the effects of vapor wall losses on SOA quantified in chamber studies and b) the effects of plume dilution (as related to fire size and meteorological stability) on SOA production. Both of these effects appear to have significant impacts on the interpretation of chamber data and ambient BB plume evolution. The scope of the work is clearly appropriate for *ACP*, the results are novel and will be of interest to many in the atmospheric chemistry community. Overall, the manuscript is well organized, the writing is good, and the presentation is clear. Thus, I recommend the paper for publication in *ACP* after the below comments are addressed.

**Specific Comments**

1. I worry that the study overestimates the effects of dilution on OA concentrations. Specifically, Fig. 6d predicts significant evaporation of OA for the two lowest-intensity fires (approximately 40% reduction in OA mass over the first ~30 min). These predictions do not seem consistent, qualitatively or quantitatively, with any ambient observations that I am aware of. For example, see Fig. 7 of Cubison et al. (2011), which compiles results for BB plume aging over similar timescales. Even in Akagi et al. (2012), where a net decrease in OA was observed, the ambient observations are qualitatively quite dissimilar from the predictions in this paper. The current results would seem to predict that BB emissions at night would undergo even more dramatic decreases in OA, since they would likely be far more impacted by dilution than chemical SOA production (even assuming nitrate radical chemistry). I'm not sure if nighttime BB plume evolution has ever been observed, and perhaps some of the differences noted above are due to fire intensity, but I would push the authors to evaluate their predictions of dilution/evaporation further.

2. As stated by the authors (line 316), OAER$_{chem}$ cannot really be evaluated against observations. It is completely dependent upon parameters that can vary quite a bit across different models. This study demonstrates a few of the model parameters that influence OAER$_{chem}$, but there are many more. I found the motivation for OAER$_{chem}$ to be quite confusing (lines 309-317). I encourage the authors to more clearly describe what it is that they hope to show with this quantity, and how it can be used in practice (beyond the current study). For example, they point to some valid limitations of OAER$_{inert}$, but there would seem to be equal (if not greater) limitations of OAER$_{chem}$ simply introduced by different models or the choice of model parameters.

3. In Section 3.4, the authors should add some discussion to prior studies that make similar observations: e.g., Capes et al. (2008) observed significant increases in O:C ratios of the organic aerosol, but a small decrease in the normalized OA mass concentrations; Hennigan et al. (2011) present similar observations through their "aged POA" analysis.

4. In the treatment of vapor wall loss, does the model allow for the reversible partitioning of vapors from the walls back to the gas phase as a compound is oxidized? Vapor wall loss is described as an equilibrium process (line 96), which implies that it is reversible – if this is/is not treated – how does this impact the current predictions?

5. This is more of a stylistic comment, but the writing in the first person is highly distracting. The terms "we" and "our" are used too extensively throughout the paper. I recommend changing to the third person voice, where possible.

6. This is probably outside the scope of this study, but it is worth noting that other factors related to fire intensity may also contribute to different aging characteristics in BB plumes (e.g., in a high intensity fire, the smoke optical thickness may produce differences in photochemistry…the formation of pyrocumulus clouds could also dramatically impact chemistry…etc.).

7. Similarly, it may be outside the scope of this study, but can the authors use their results to make conclusions about the evolution of BB emissions at night?

8. Finally, the References need to be carefully checked – they are out of order, and some are not the correct form (e.g., ACPD article cited when the article has been published in ACP).

**Technical Corrections**

1. Delete Lines 141 – 149 ("We describe our aerosol microphysics model…presents our conclusions.") – the sections have clear headings so this is redundant.

2. Delete the sentence starting on line 320 – the section heading is just above this sentence.

3. Delete the sentence starting on line 344 – the section heading is just above this sentence.

4. Line 571: change "the" to "some"

5. Line 67: Grieshop et al. (2009) was a chamber study, not a field study.

6. Line 188: most chambers are rectangular or cubic – what is chamber radius?

7. Line 324: rewrite this sentence to be less awkward.

8. Line 328: "…simulations are shown…"

9. Line 438: is the term "perfect accommodation" technically preferred?

10. Line 520: do the authors mean 'OA' instead of 'BC'?

**References**

Akagi, S. K., et al.: Evolution of trace gases and particles emitted by a chaparral fire in California, Atmos. Chem. Phys., 12, 1397-1421, 2012.

Capes, G., et al.: Aging of biomass burning aerosols over West Africa: Aircraft measurements of chemical composition, microphysical properties, and emission ratios, J. Geophys. Res., 113, D00C15, 2008.

Cubison, M. J., et al.: Effects of aging on organic aerosol from open biomass burning smoke in aircraft and laboratory studies, Atmos. Chem. Phys., 11, 12049–12064, doi:10.5194/acp-11-12049-2011, 2011.

Grieshop, A. P., et al.: Laboratory investigation of photo chemical oxidation of organic aerosol from wood fires 1: measurement and simulation of organic aerosol evolution, Atmos. Chem. Phys., 9, 1263-1277, doi:10.5194/acp-9-1263-2009, 2009

Hennigan, C. J., et al.: Chemical and physical transformations of organic aerosol from the photooxidation of open biomass burning emissions in an environmental chamber, Atmos. Chem. Phys., 11, 7669-7686, 2011

---

## Referee Comment (RC2) · Anonymous Referee #2 · 7 Jan 2017

Bian et al. investigate processes governing biomass-burning organic aerosol evolution in smog chamber and ambient plumes with the use of an aerosol microphysics model. An extensive set of sensitivity analysis is performed to evaluate the role of vapor wall loss, file size, dilution, and atmospheric stability in the SOA formation and evolution. Overall, the topic is very important and the manuscript is well written. One concern is the extent to which the simulations could adequately represent the actual measurements in chamber- and/or field-derived biomass-burning plumes. Specific comments are listed below that I would like the authors to address (consider) before publication in ACP.

General:

The authors found that accounting for vapor wall losses leads to 2-3 times increases in the total SOA production in chamber experiments. This conclusion, however, depends on how the adjustable parameters in the model are tuned against chamber observations, and as a result the enhancement in SOA production in the absence of vapor wall loss could vary with different model parameterizations. The authors mentioned that the OA concentrations from FLAME-III experiments are used to constrain the model performance, yet the comparison of simulations with experimental observations are not given in details throughout of the paper. Ideally, the organic aerosol temporal profile by AMS/SMPS during one representative experiment should be given together with corresponding simulations (e.g., Figure 3) to better visualize the model performance. Another question related, have the authors conducted optimal fitting of simulations to chamber measured quantities such as organic aerosol mass, O:C and H:C ratios? Is there more than one set of parameters that could well represent the observations? What is the physical meaning of each best-fit parameter that is chosen to describe the BBOA evolution?

For the ambient plume simulations, the authors are suggested to add discussions on how the values of key parameters, such as fire sizes and atmospheric stability classes, are assigned. Are they representative of the fire plume transportation in the air? A thorough search on the ambient fire plume properties in literatures might be useful to rationalize the sensitivity tests conducted in this study.

Recent two-dimensional VBS frameworks have incorporated gas-phase fragmentation processes as a function of the O:C ratio of individual volatility bins (e.g., Jimenez et al. 2009). The original distribution of volatility bins upon one generation of oxidation (drops in volatility per reactions) would correspondingly change by adding this branch of mechanism into the model framework. Upon OH-exposure in the order of ~ $10^{10}$ molecules $cm^{-3}$ s (typically several hours of reactions in the atmosphere), fragmentation should have occurred to some extent, depending on the OH reactivity of the parent precursors. The authors are suggested to discuss uncertainties caused by the assumption of zero fragmentation in the reaction mechanisms.

Minor:

Page 5, Line 168: How are the black carbon and organic contents treated in each particle size bin in the model? Are they well mixed?

Page 9, Line 328: Add 'are' before 'shown'.

Page 29, Figure 1: How are the vapor concentrations calculated, based on equilibrium partitioning?

---

## Author Comment (AC1) · 21 Jan 2017

We thank reviewer 1 for their thoughtful and helpful review.  Our response is below.

1. *I worry that the study overestimates the effects of dilution on OA concentrations. Specifically, Fig. 6d predicts significant evaporation of OA for the two lowest-intensity fires (approximately 40% reduction in OA mass over the first ~30 min). These predictions do not seem consistent, qualitatively or quantitatively, with any ambient observations that I am aware of. For example, see Fig. 7 of Cubison et al. (2011), which compiles results for BB plume aging over similar timescales. Even in Akagi et al. (2012), where a net decrease in OA was observed, the ambient observations are qualitatively quite dissimilar from the predictions in this paper.*

Response: Our simulations suggest the patterns of OA evolution are sensitive to the fire sizes. The burn area for Williams prescribed fire in the study of Akagi et al. (2012) was 81 hectare (i.e. 0.81 km$^2$). This may be more comparable with our simulation for the fire size of 1 km$^2$. May et al. (2015) showed that OA reduction was approximately 50% during the plume evolution, which is more similar to our smaller simulated fires. We added the following in the main text from line 68: "…production or even a net loss (Akagi et al., 2012: May et al., 2015). OA loss in first hour after emission was approximately 50% in the study of May et al. (2015), OA consists of …"

*The current results would seem to predict that BB emissions at night would undergo even more dramatic decreases in OA, since they would likely be far more impacted by dilution than chemical SOA production (even assuming nitrate radical chemistry). I'm not sure if nighttime BB plume evolution has ever been observed, and perhaps some of the differences noted above are due to fire intensity, but I would push the authors to evaluate their predictions of dilution/evaporation further.*

7. *Similarly, it may be outside the scope of this study, but can the authors use their results to make conclusions about the evolution of BB emissions at night?*

We combined our response to address above two points about night-time evolution: It may be difficult to generalize about day/night differences due to various aspects being different between day and night on average. In general, nighttime plumes may have (1) less dispersion in the boundary layer due to more-stable air, (2) different chemistry, and (3) lower emission fluxes as peak fire intensities are typical during the day (this may affect fire size too; Zhang and Kondragunta, 2008; Wooster and Lagoudakis, 2009). It's unclear how the convolution of these differences might impact the plumes, and it probably varies between cases.

We added the text from line 554: "For nighttime OA evolution, it may be difficult to generalize about day/night differences due to various aspects being different between day and night on average. In general, nighttime plumes may have (1) less dispersion in

the boundary layer due to more-stable air, (2) different chemistry, and (3) lower emission fluxes as peak fire intensities are typical during the day (this may affect fire size too; Zhang and Kondragunta, 2008; Wooster and Lagoudakis, 2009). It's unclear how the convolution of these differences might impact the plumes, and it probably varies between cases."

2. *As stated by the authors (line 316), OAERchem cannot really be evaluated against observations. It is completely dependent upon parameters that can vary quite a bit across different models. This study demonstrates a few of the model parameters that influence OAERchem, but there are many more. I found the motivation for OAERchem to be quite confusing (lines 309-317). I encourage the authors to more clearly describe what it is that they hope to show with this quantity, and how it can be used in practice (beyond the current study). For example, they point to some valid limitations of OAERinert, but there would seem to be equal (if not greater) limitations of OAERchem simply introduced by different models or the choice of model parameters.*

Response: $OAER_{chem}$ can certainly vary across models due to different assumptions. We introduced $OAER_{chem}$ to isolate the effect of SOA formation to give an alternate metric to $OAER_{inert}$, which is the convolution of evaporation and SOA formation. While $OAER_{chem}$ cannot be evaluated against measurements, it does tell us what the isolated impact of SOA formation is *for the choice of model parameters used in the simulation*. We have modified the motivation of $OAER_{chem}$ from line 309: "To isolate the impact of SOA formation alone on our simulations, we introduce the chemistry OA mass enhancement ratio ($OAER_{chem}$) to give an alternate metric of $OAER_{inert}$ (which is the convolution of both evaporation and SOA formation). We define $OAER_{chem}$ as the ratio of predicted…"

3. *In Section 3.4, the authors should add some discussion to prior studies that make similar observations: e.g., Capes et al. (2008) observed significant increases in O:C ratios of the organic aerosol, but a small decrease in the normalized OA mass concentrations; Hennigan et al. (2011) present similar observations through their "aged POA" analysis.*

Response: We have added text in the lines 552-555. "Papers analyzing field observations have suggested this possibility. Capes et al. (2008) and Cubison et al. (2011) observed significant increases in O:C ratios of the organic aerosol with aging, but a small decrease in the normalized OA mass concentrations; Akagi et al. (2012) observed the decrease of OA with aging and attributed this to the processes of particle evaporation. Similarly, Jolleys et al. (2015) observed increasing O:C elemental ratio with aging but lowering normalized OA concentrations in the smoke plumes, and they attributed this to the combination of dilution and chemical processing. May et al. (2015)

also suggested the competition between dilution-driven evaporation and SOA formation during the plume transport may be occurring in their observed plumes, as they found approximately 50% reduction of OA after several hours of aging with increasing in the O:C ratio. Additionally, the lab study of Hennigan et al. (2011) also showed increased O:C ratios in experiments with decreasing OA concentrations. Our modeling result is consistent with the findings from these observational studies reporting increased oxygenation with time for the OA even with observed decreases in the relative amount of OA (or a relative constant or lower $OAER_{inert}$).”

4. *In the treatment of vapor wall loss, does the model allow for the reversible partitioning of vapors from the walls back to the gas phase as a compound is oxidized? Vapor wall loss is described as an equilibrium process (line 96), which implies that it is reversible – if this is/is not treated – how does this impact the current predictions?*

Response: The vapor wall loss is treated as a reversible partitioning process. Previous studies (Bian et al., 2015, Zhang et al, 2015) suggested two variables could influence vapor wall loss: the effective saturation of vapor with respect to the wall ($C_w/M_w\gamma_w$) and the accommodation coefficient for vapor into the wall ($\alpha_w$). We performed the sensitivity tests on these two variables and showed that the simulations overlap with the measurement of Hennigan et al. (2011). However, as we stated in the manuscript, we are unable to determine which set of $\alpha_w$, $C_w/M_w\gamma_w$, and chemistry assumptions best represent the actual processes occurring in the chamber, since different combinations of these values can reproduce the observed $OAER_{inert}$ range. However, if vapor wall loss is turned off, the amount of OA mass increases greatly over simulations with vapor wall loss on – regardless of what vapor-wall-loss and chemistry parameters are chosen. Therefore, the prediction of vapor wall loss has large uncertainties depending on the two variables, but this does not influence our main conclusion.

5. *This is more of a stylistic comment, but the writing in the first person is highly distracting. The terms "we" and "our" are used too extensively throughout the paper. I recommend changing to the third person voice, where possible.*

Our use of "we" and "our" is to keep our writing concise and direct, and to generally use the active voice. This link (https://cgi.duke.edu/web/sciwriting/index.php?action=passive_voice) provides a nice overview of the pros and cons of active and passive voice in scientific writing (but does not conclude that one must err to using one or the other). I (Jeff Pierce writing here) feel personally that the advantages of active voices outway disadvantages, and I personally find writing that avoids "we" and "our" harder to follow and more work to read. If you see me (Jeff again) at a conference or meeting, feel free to approach me about this if you don't mind losing your anonymity. I'm interested in learning about why you

feel "we" and "our" is distracting as I realize that not everyone has the same writing preferences, and it's good to try to write in a way that satisfies as broad an audience as possible.

6. *This is probably outside the scope of this study, but it is worth noting that other factors related to fire intensity may also contribute to different aging characteristics in BB plumes (e.g., in a high intensity fire, the smoke optical thickness may produce differences in photochemistry…the formation of pyrocumulus clouds could also dramatically impact chemistry…etc.).*

Fire intensity certainly influences OA evolution in the plume. We performed the test on the high and low emission mass flux ($5 \times 10^{-6}$ and $2 \times 10^{-8}$ kg m$^{-2}$s$^{-1}$). For OA evolution for fire size of 1 km$^2$ under Atmospheric Class of D in Fig 7 and 8, high emission mass flux (i.e. large fire intensity) has lower OAER$_{inert}$ and OAER$_{chem}$, compared with low emission mass flux, suggesting that under the same dilution ratio, lower emission mass flux has slightly more-effective SOA formation. OA concentrations for lower emission mass flux quickly drop close or below the background non-volatile OA concentrations and further dilution does not lead to further evaporation. The evaporated organics are available for SOA formation. Both of OAER$_{inert}$ and OAER$_{chem}$ after 4 hrs were thus higher for low emission mass flux than high emission mass flux.

We added text to the paragraph after line 523: "Fire intensity also influences OA evolution in the plume through changes in emission fluxes. Compared OA evolution for fire size of 1 km$^2$ under Atmospheric Class of D in Fig 7 and 8, high emission mass flux (i.e. large fire intensity) has lower OAER$_{inert}$ and OAER$_{chem}$ than that of low emission mass flux, suggesting that under the same dilution ratio, lower emission mass flux has slightly more-effective SOA formation. OA concentrations for lower emission mass flux quickly drop close or below the background non-volatile OA concentrations and further dilution does not lead to further evaporation. The evaporated organics are available for SOA formation. Both of OAER$_{inert}$ and OAER$_{chem}$ after 4 hrs were thus higher for low emission mass flux than high emission mass flux." We also added the background OA concentration line in the Figures 6 to 8.

Mok et al. (2016) estimated that the reduced UV due to brown carbon and black carbon could slow down the photochemical rate as radicals OH, HO2 and RO2 was decreased in the plume by 17%, 15% and 14%, respectively. Also, cloud processing of smoke from biomass burning in the pyrocumulus clouds (and other clouds that the plume cycles through) could largely alter the smoke chemistry (Yokelson et al., 2003; Akagi et al., 2011). However, due to the limited information to constrain the chemical mechanism in our model, we only simulate gas-phase functionalization and do not include aerosol-phase or heterogeneous reactions, or cloud processing. We also added the text after

line 228: "We also do not include aerosol-phase or heterogeneous reactions, cloud processing, or effects of smoke on oxidant fields in our model, although these processes may  affect the chemistry of plume (Yokelson et al., 2003; Akagi et al., 2011; Mok et al., 2016). The SOA mass yield $\alpha_{i,j}$ is assumed to be 1 for all reactions. We use this simple assumption of chemistry as a first test in our chamber and plume systems as we found that we did not have enough information to constrain gas-phase yields or additional chemistry mechanisms beyond this."

8. *Finally, the References need to be carefully checked – they are out of order, and some are not the correct form (e.g., ACPD article cited when the article has been published in ACP).*

Corrected.

**Technical Corrections**

1. *Delete Lines 141 – 149 ("We describe our aerosol microphysics model…presents our conclusions.") – the sections have clear headings so this is redundant.*

Done.

2. *Delete the sentence starting on line 320 – the section heading is just above this sentence.*

Done.

3. *Delete the sentence starting on line 344 – the section heading is just above this sentence.*

Done.

4. *Line 571: change "the" to "some"*

Done.

5. *Line 67: Grieshop et al. (2009) was a chamber study, not a field study.*

Deleted.

6. *Line 188: most chambers are rectangular or cubic – what is chamber radius?*

The chamber of Carnegie Mellon University was nearly cubic. We assume the chamber to be a sphere to allow for an analytical solution of turbulent wall-loss rates following Crump and Seinfeld (1981) and implemented in Pierce et al., (2008) on a similarly shaped chamber.

Changed to "…$R$ is the radius of the chamber on the assumption that the chamber is a sphere…"

7. *Line 324: rewrite this sentence to be less awkward.*

Done.

8. *Line 328: "…simulations are shown…"*

Done.

9. *Line 438: is the term "perfect accommodation" technically preferred?*

Changed to "…A value of 1 represents no limitation on the vapor-wall loss rates due to this process…"

10. *Line 520: do the authors mean 'OA' instead of 'BC'?*

Corrected.

References:

Akagi, S. K., et al.: Emission factors for open and domestic biomass burning for use in atmospheric models, Atmos. Chem. Phys., 11, 4039-4072, 2011

Akagi, S. K., et al.: Evolution of trace gases and particles emitted by a chaparral fire in California, Atmos. Chem. Phys., 12, 1397-1421, 2012.

Bian, Q., et al.: Investigation of particle and vapor wall-loss effects on controlled wood-smoke smog-chamber experiments, Atmos. Chem. Phys., 15, 11027-11045, 2015

Capes, G., et al.: Aging of biomass burning aerosols over West Africa: Aircraft measurements of chemical composition, microphysical properties, and emission ratios, J. Geophys. Res., 113, D00C15, 2008.

Crump and Seinfeld: Turbulent deposition and gravitational sedimentation of an aerosol in a vessel of arbitrary shape, J Aerosol Sci., 5, 405-415, 1981.

Cubison, M. J., et al.: Effects of aging on organic aerosol from open biomass burning smoke in aircraft and laboratory studies, Atmos. Chem. Phys., 11, 12049-12064, 2011.

May, A. A., et al: Observations and analysis of organic aerosol evolution in some prescribed fire smoke plumes, Atmos. Chem. Phys., 15, 6323-6335, 2015.

Mok, J., et al.: Impacts of brown carbon from biomass burning on surface UV and ozone photochemistry in the Amazon Basis, Scientific reports 6, 36940, doi:10.1038/srep36940, 2016.

Roberts, G and Lagoudakis, E.: Annual and diurnal African biomass burning temporal dynamics, Biogeosciences, 6, 849-866, 2009.

Yokelson, R. J., et al.: Trace gas measurements in nascent, aged, and cloud-processed smoke from African savanna fires by airborne Fourier transform infrared spectroscopy (AFTIR), J. Geophys. Res., 108, 8478, doi: 10. 1029/2002JD002322, D13, 2003

Zhang, X. and Kondragunta, S.: Temporal and spatial variability in biomass burned areas across the USA derived from the GOES fire product, Remote Sens. Environ., 112, 2886-2897, 2008.

---

## Author Comment (AC2) · 21 Jan 2017

We thank reviewer 2 for their thoughtful and helpful review. Our responses are below.

*The authors found that accounting for vapor wall losses leads to 2-3 times increases in the total SOA production in chamber experiments. This conclusion, however, depends on how the adjustable parameters in the model are tuned against chamber observations, and as a result the enhancement in SOA production in the absence of vapor wall loss could vary with different model parameterizations. The authors mentioned that the OA concentrations from FLAME-III experiments are used to constrain the model performance, yet the comparison of simulations with experimental observations are not given in details throughout of the paper. Ideally, the organic aerosol temporal profile by AMS/SMPS during one representative experiment should be given together with corresponding simulations (e.g., Figure 3) to better visualize the model performance.*

Response: We do not compare each individual simulation with each corresponding observation because May et al. (2015) was only able to derive a single volatility distribution across the FLAME III 18 experiments. Additionally the IVOC volatility distribution that we use was from FLAME IV experiments that do not directly correspond to the specific FLAME III experiments. Thus, we do expect error in individual experiments due to these assumptions, but we seek to capture the mean behavior across all of the experiments.

We added the following text near Line 408: "…are in very good agreement with those observations. May et al. (2015) was only able to derive a single volatility distribution across the FLAME III 18 experiments and the IVOC volatility distribution from FLAME IV experiments do not directly correspond to the specific FLAME III experiments. Thus, we expect the error in individual experiments due to the single volatility distributions across all simulations. We thus seek to capture the mean behavior across all of the experiments rather than comparing individual simulations to their corresponding experiments. Our simulations also show that…"

*Another question related, have the authors conducted optimal fitting of simulations to chamber measured quantities such as organic aerosol mass, O:C and H:C ratios?*

We do not simulate O:C and H:C. We do compare organic aerosol mass, at least implicitly; this is what is being evaluated in Figures 4 and 5. The initial organic aerosol masses in the simulations are identical to the measurements, $OAER_{inert}$ effectively evaluates OA mass as the inert-tracer wall losses for these experiments have been evaluated in Bian et al., 2015.

We added the following text around Line 406: "…Since the initial organic aerosol masses in the simulations are identical to the measurements, we use $OAER_{inert}$ to

evaluate simulated OA mass against measurements in Figs. 4 and 5  as the inert-tracer wall losses for these experiments have been evaluated in Bian et al. (2015)."

*Is there more than one set of parameters that could well represent the observations? What is the physical meaning of each best-fit parameter that is chosen to describe the BBOA evolution?*

Yes, more than one set of parameters could well represent the observations because we do not have enough experimental data to determine which combination of parameters for the study of FLAME III. This is shown in Figures 4 and 5, and we discuss the conclusion that multiple sets of assumptions can describe the measurements from line 415 to 456. We neither evaluate nor declare "best fit parameters" for the study of the influence of wall loss on secondary-organics evolution.

To reinforce this point, we modified the text from line 453, "…can better represent the FLAME-III experiments; however, we are unable to provide the "best-fit parameters" for the simulations as we cannot determine which set of $\alpha_w$, $C_w/M_w\gamma_w$, and chemistry assumptions best represent the actual processes occurring in the chamber…"

*For the ambient plume simulations, the authors are suggested to add discussions on how the values of key parameters, such as fire sizes and atmospheric stability classes, are assigned. Are they representative of the fire plume transportation in the air? A thorough search on the ambient fire plume properties in literatures might be useful to rationalize the sensitivity tests conducted in this study.*

Cochrane et al. (2012) reported 14 wildfires with fire size from 5.28 to 1868.78 $km^2$. The burning areas for prescribed fires observed in Akagi et al. (2013) ranged from 0.162 to 1.47 $km^2$. The fire size for agricultural and pile burns can be as small as $7\times10^{-5}$ $km^2$ (Springsteen, et al. 2015). The fire sizes in our sensitive test are $10^{-4}$, 0.01, 1, 100 $km^2$, covering most of this observed range. Atmospheric stability includes six classes from A (unstable) to F(stable), which represent all the possible atmospheric stability conditions that range from clear sunny days (very unstable) to calm clear nights (very stable). We do not try to simulate any specific fire, just a range of possible conditions. We revised the manuscript from lines 477 to 479 accordingly:

"…The initial plume width is associated with fire size, which means that the fire size could largely influence the plume evolution (Sakamoto et al., 2016). Cochrane et al. (2012) reported 14 wildfires with fire size from 5 to over 1000 $km^2$. Akagi et al. (2013) also recorded the burn areas for the observed prescribed fire range from 0.162 to 1.47 $km^2$. The burning area for Williams fire was 0.81 $km^2$ (Akagi et al., 2012). The fire size for agricultural and pile burns can be as small as $7\times10^{-5}$ $km^2$ (Springsteen, et al. 2015). We therefore perform simulations on the evolution of ambient OA concentrations over 4

hours of simulated transport, for four different fire areas of $1\times10^{-4}$, $1\times10^{-2}$, $1\times10^{0}$ and $1\times10^{2}$ km$^2$ (with the fire width assumed to be the square root of these areas), which largely cover the reported burned areas above…"

*Recent two-dimensional VBS frameworks have incorporated gas-phase fragmentation processes as a function of the O:C ratio of individual volatility bins (e.g., Jimenez et al. 2009). The original distribution of volatility bins upon one generation of oxidation (drops in volatility per reactions) would correspondingly change by adding this branch of mechanism into the model framework. Upon OH-exposure in the order of ~ $10^{10}$ molecules cm$^{-3}$ s (typically several hours of reactions in the atmosphere), fragmentation should have occurred to some extent, depending on the OH reactivity of the parent precursors. The authors are suggested to discuss uncertainties caused by the assumption of zero fragmentation in the reaction mechanisms.*

We agree that lack of simulating fragmentation is a limitation of our study. We have added text to emphasize this after line 341: "…and the alkane $k_{OH}$ set with the two-volatility-bin drop per reaction as a lower bound for SOA formation. Jimenez et al. (2009) showed that fragmentation would produce more-volatile species compared with parent species. The assumption of zero fragmentation and unity SOA mass yield may cause overestimation of SOA production in our study."

We also changed the text from Line 581: "…Uncertainties in parameters that control vapor wall losses, such as the wall saturation concentration and wall accommodation coefficient, as well as uncertainties in gas-phase chemistry with the assumption of zero fragmentation and unity SOA mass yield, lead to uncertainties in our simulations."

*Minor:*

*Page 5, Line 168: How are the black carbon and organic contents treated in each particle size bin in the model? Are they well mixed?*

We assume that all species are internally mixed within each size section, meaning the black and OA exists in the all particles at the same ratio within each size bin. However, for purposes for calculating OA partitioning, we assume that OA and black carbon exist in separate phases within each particle, and thus this presence of black carbon does not enhance partitioning of OA to the particle phase. We have added text after Line 168: "… and water with 36 logarithmically spaced size sections from 3 nm to 10 μm. We assume that all species are internally mixed within each size section, meaning that the ratio of BC and OA are the same for all particles within each size bin. When calculating OA partitioning, we assume that OA and BC exist in separate phases, and thus the presence of BC does not influence OA partitioning to the particle phase in the model. In our previous study examining the influence of wall loss…"

*Page 9, Line 328: Add 'are' before 'shown'.*

Corrected.

*Page 29, Figure 1: How are the vapor concentrations calculated, based on equilibrium partitioning?*

We assume that you are asking how the initial vapor concentrations are calculated. We set the organic aerosol concentration equal to that measured by the AMS, assume the total-organic volatility distributions from May et al. (2015) and Hatch et al. (2016), and estimate the vapor concentration necessary to sustain the AMS-measured OA mass based on aerosol partitioning theory (Pankow, 1994) on the assumption of gas and particle equilibrium partitioning.

We revised lines 175 to 179 : "In this current study, we expand the simulated organics from eight to fifteen "species" including more volatile organics between $10^6$ to $10^{11}$ µg m$^{-3}$, based on the FLAME-4 study of Hatch et al. (2016), to account for chemical transformations from both volatile and semivolatile organic species and estimate the initial organic vapor concentration based on aerosol partitioning theory (Pankow, 1994) on the assumption of gas and particle equilibrium partitioning (Fig. 1a). The evolution of the organic vapors is calculated based on partitioning theory (to get equilibrium vapor pressures above the particle), wall-equilibrium vapor pressures, and kinetic mass transfer to/from the particles and the walls. "

References:

Akagi, et al.: Evolution of trace gases and particles emitted by a chaparral fire in California, Atmos. Chem. Phys., 12, 1397-1421, 2012

Akagi, et al.: Measurements of reactive trace gases and variable $O_3$ formation rates in some South Carolina biomass burning plumes, Atmos. Chem. Phys., 13, 1141-1165, 2013.

Cochrane, et al: Estimation of wildfire size and risk changes due to fuels treatments, Int. J. Wildland Fire, 21, 357-367, 2012

Jimenez, et al.: Evolution of Organic Aerosols in the Atmosphere, Science 326, 1525, doi:10.1126/science.1180353, 2009

Springsteen, et al.: Forest biomass diversion in the Sierra Nevada: Energy, economics and emissions, Calif. Agric., 69, 142-149, DOI: 10.3733/ca.v069n03p142, 2015

Pankow, J. F.: An absorption model of gas/particle partitioning of organic compounds in

the atmosphere, Atmos. Environ., 28, 185–188, 1994.

Yokelson, et al.: Emissions from biomass burning in the Yucatan, Atmos. Chem. Phys., 9, 5785-5812, 2009.